# Fine-Tuning Large Vision-Language Models as Decision-Making Agents via Reinforcement Learning

**Yuexiang Zhai**[1*]  **Hao Bai**[2†]  **Zipeng Lin**[1†]  **Jiayi Pan**[1†]  **Shengbang Tong**[3†]  **Yifei Zhou**[1†]

**Alane Suhr**[1]  **Saining Xie**[3]  **Yann LeCun**[3]  **Yi Ma**[1]  **Sergey Levine**[1]

[1]UC Berkeley  [2]UIUC  [3]NYU

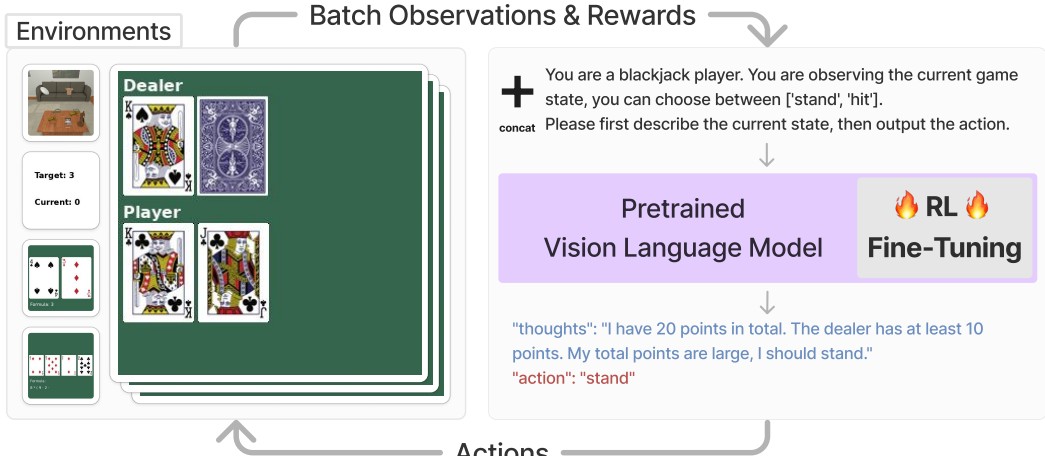

Figure 1: **Method overview.** We propose a framework for training large Vision-Language Models (VLM) with Reinforcement Learning (RL). At each time step, the VLM takes the current observation and a predesigned prompt as input and outputs an utterance containing a chain of thought reasoning and a text action. The text action is parsed into the environment for generating task rewards. Finally, we apply RL with the task reward to fine-tune the entire VLM.

## Abstract

Large vision-language models (VLMs) fine-tuned on specialized visual instruction-following data have exhibited impressive language reasoning capabilities across various scenarios. However, this fine-tuning paradigm may not be able to efficiently learn optimal decision-making agents in multi-step goal-directed tasks from interactive environments. To address this challenge, we propose an algorithmic framework that fine-tunes VLMs with reinforcement learning (RL). Specifically, our framework provides a task description and then prompts the VLM to generate chain-of-thought (CoT) reasoning, enabling the VLM to efficiently explore intermediate reasoning steps that lead to the final text-based action. Next, the open-ended text output is parsed into an executable action to interact with the environment to obtain goal-directed task rewards. Finally, our framework uses these task rewards to fine-tune the entire VLM with RL. Empirically, we demonstrate that our proposed framework enhances the decision-making capabilities of VLM

---

*Project Lead, email: simonzhai@berkeley.edu. Project page: https://rl4vlm.github.io/
†Equal contribution, listed in alphabetical order, see Appendix A for list of contributions.

38th Conference on Neural Information Processing Systems (NeurIPS 2024).

agents across various tasks, enabling 7b models to outperform commercial models such as GPT4-V or Gemini. Furthermore, we find that CoT reasoning is a crucial component for performance improvement, as removing the CoT reasoning results in a significant decrease in the overall performance of our method.

# 1 Introduction

Large vision-language models (VLMs) [7, 44, 18] demonstrate remarkable capabilities as general-purpose agents in solving various tasks through language reasoning. In particular, fine-tuning VLMs with specialized visual instruction following data appears to be a key technique for improving the capabilities of VLMs [34, 84, 33, 30]. However, visual instruction tuning may not be optimal for training decision-making agents in multi-step interactive environments requiring visual recognition and language understanding, as visual instruction tuning mainly performs supervised learning on pre-collected datasets without interacting with the environments [22]. Consequently, if the pre-collected datasets lack sufficient diversity to cover a wide range of decision-making scenarios, visual instruction tuning may fail to improve the VLM agent's decision-making capabilities.

To unleash the learning capabilities of VLM agents in multi-step goal-directed decision-making environments, reinforcement learning (RL), a method that has proven effective in training multi-step interactive agents [41, 59, 6, 69], naturally offers a paradigm that supports this purpose. However, while RL has been widely adopted for training purely text-based tasks for large language models (LLMs) [60, 50, 1, 83], end-to-end VLM fine-tuning with RL for goal-directed multi-step tasks has not yet been studied, to the best of our knowledge.

Our main contribution in this paper is an algorithmic framework that directly fine-tunes VLMs with RL for multi-step goal-directed decision-making tasks requiring vision-language understanding. In our framework, the VLM first receives a task description prompt, which guides it to generate task-specific chain-of-thought (CoT) reasoning [75, 73] (blue parts in Figure 1), followed by a text-based action (red parts in Figure 1). The CoT reasoning is designed for efficient explorations by prompting the VLMs to generate intermediate reasoning that leads to the final text-based action. Our framework then parses the text-based actions into executable actions for the environment, which generates potentially goal-directed rewards and the next state for RL training.

To evaluate the effectiveness of our method in enhancing a VLM's decision-making capabilities, we adopt a 7b model [35] as the backbone VLM and apply our method to five decision-making tasks. These tasks come from two domains: an original domain, which evaluates the VLM's decision-making capabilities requiring fine-grained visual recognition and language reasoning, and an embodied AI domain [58] focusing on testing tasks demanding visual semantic reasoning capabilities. Empirical results show that our method enhances the decision-making capabilities of VLMs in both domains, enabling 7b models to surpass the performance of commercial models such as GPT4-V [44] and Gemini [18]. Moreover, our experiments reveal that CoT reasoning is crucial for performance improvement in our RL training. Specifically, we test our method on the same tasks *without* the CoT reasoning and observe a significant drop in overall performance in both domains.

# 2 Related Work

**Training LLMs or VLMs with RL.** RL has been widely adopted for training LLMs and VLMs [85, 61, 70, 45, 10, 50, 9, 43, 18, 62, 60, 1, 20, 83]. Some studies [85, 61, 45, 10, 43, 18, 62] focus on applying RL from human feedback (RLHF), which involves learning reward models from human feedback before deploying RL. Other research [50, 9, 60, 1, 20, 83] focuses on deploying RL with task-specific reward functions without using human preference data. Our paper is similar to the latter [50, 9, 60, 1, 20, 83] which applies RL to train LLMs on customized reward functions from different environments. There are two major differences between our paper and prior works [50, 60, 1, 20, 83]. Firstly, our method incorporates visual inputs, broadening its applicability to a wider range of tasks that require vision-language understanding or multimodal reasoning [29, 38]. Secondly, while previous works do not explore how CoT reasoning affects RL training on large models in general, we identify CoT reasoning as a crucial component for enhancing RL training. We empirically observe that incorporating CoT reasoning significantly improves the overall performance of RL training on *all* tested domains.

**Adopting LLMs and VLMs as decision-making agents.** Many prior works have studied various methods of using frozen LLMs and VLMs for decision-making. One line of work studies the prompting techniques [75, 14, 79, 78, 74, 31, 76, 47, 71, 48, 24] for enhancing the decision-making capabilities of large foundation models, see Dong et al. [14], Yang et al. [77] for a detailed survey for other prompting based methods. Our work differs from all prompting-based methods since we directly use RL to fine-tune the entire VLM as decision-making agents. Other studies [42, 64, 4, 52, 11] integrate frozen VLMs ot LLMs into their training pipeline for processing task descriptions or feature extraction, without using text-based actions. focuses on integrating different components from VLMs for downstream RL training. For example, some studies use the VLMs or CLIP vision encoder [46, 42, 64] as reward models for training, which differs from our method since we adopt rewards from the environments. Other studies [42, 64, 11] integrate frozen VLMs/LLMs into their training pipeline for processing task descriptions [42, 64, 46] or feature extraction [11], without using text-based actions. Our paper differs from these works [42, 64, 11] in two major aspects. From a technical perspective, we focus on a more challenging paradigm by directly fine-tuning the entire VLM with RL, whereas previous methods [42, 64, 11] only train additional MLP or transformer layers to connect the frozen LLM/VLM with the action space. More importantly, our method directly interacts with the environments using *open-ended text*, enabling it to utilize the CoT reasoning capability of VLMs for more efficient explorations for decision-making.

**Evaluating VLMs as decision-making agents.** Previous studies have thoroughly examined the fundamental evaluations of VLMs in non-interactive tasks [3, 37, 80, 32, 65, 81, 16]. Our focus, however, is on evaluating a VLM's decision-making capabilities in interactive environments that require both visual recognition and language reasoning. Representative interactive environments include purely text-based environments [13, 28, 72] or embodied AI environments [40, 58, 56, 15]. We adopt the ALFWorld [58] embodied environment for evaluating our method's ability to improve VLM's visual semantic reasoning capabilities. In addition to the ALFWorld embodied AI environment, we also design an original "gym-like" [8] environment to test VLM's decision-making capabilities in tasks that require fine-grained visual recognition and language reasoning.

**CoT prompting.** Recent studies in prompting for LLMs have demonstrated the crucial role of CoT in enhancing complex reasoning capabilities [75, 26, 17, 73, 82, 79]. Wei et al. [75] show that CoT reasoning can significantly boost LLMs' performance across different reasoning tasks by showing that adding simple exemplar-based prompts, leading to better performance on benchmarks such as the GSM8K [12]. A follow-up study [73] proposes a novel self-consistency decoding strategy that explores multiple reasoning paths, demonstrating substantial gains in arithmetic and commonsense reasoning tasks. Other works [26, 82, 17] have shown that adding prompts to break complex tasks into subtasks and solve them step-by-step significantly improves LLM's reasoning capability. Our work differs from these CoT prompting studies as we aim to provide an algorithmic framework that can train VLMs with RL, where the CoT prompting appears as a key component of the framework. In contrast, prior works focus on improving the reasoning capabilities of LLMs with increasingly sophisticated prompting of frozen models.

## 3 Preliminaries

**Standard RL terminologies.** We follow the standard notations from classic RL literature [63, 2]. Specifically, we use $\mathcal{M} = \{\mathcal{S}, \mathcal{A}, P, r, \gamma\}$ to denote an MDP, where $\mathcal{S}$ denotes the state space, $\mathcal{A}$ denotes the action space, $P$ denotes the transition dynamics, $r : \mathcal{S} \times \mathcal{A} \to \mathbb{R}$ denotes the reward function and $\gamma \in [0, 1]$ denotes the discount factor. Our goal is to learn a policy $\pi : \mathcal{S} \to \mathcal{A}$ that maximizes the overall discounted return $\max_{\pi \in \Pi} \mathbb{E}_\pi \left[ \sum_{t=0}^{T} \gamma^t r(s_t, a_t) \right]$, where $T$ is the maximum number of steps per episode. Without loss of generality, we use $\pi(a|s) \in [0, 1]$ to denote probability of $\pi$ choosing $a$ at $s$.

**Adapting the RL formalism to VLMs.** We use $\mathcal{V}$ to denote the discrete and finite vocabulary (token) space, and we use $\mathcal{V}^m, \mathcal{V}^n$ to represent the input and output text space, where $m$ and $n$ represent the maximum token length of the input and output sequence. We adapt the RL formalism to VLMs by treating the combination of the *vision and language inputs* to VLMs as the state space: $\mathcal{S} = \mathcal{O} \times \mathcal{V}^m$, where $\mathcal{O}$ is the space of all RGB images. We view each utterance [1, 83] of the

language outputs from VLMs as the action space $\mathcal{V}^n$. Therefore, the input and output of a VLM policy with parameter $\theta$ can be written as $\pi_\theta : \mathcal{O} \times \mathcal{V}^m \to \mathcal{V}^n$. For example, in the `Blackjack` task shown in Figure 1, each state $s$ consists of an RGB image $o$ with the cards of the dealer and the player, as well as an input prompt $\boldsymbol{v}^{\text{in}}$ with maximum token length $m$, and the text output $\boldsymbol{v}^{\text{out}} = \pi_\theta(o, \boldsymbol{v}^{\text{in}})$ (with a maximum token $n$) will later be parsed as an action to interact with the environment. Similar to the standard RL setting, we use $\pi_\theta(\boldsymbol{v}^{\text{out}}|o, \boldsymbol{v}^{\text{in}}) \in [0, 1]$ to denote the probability of a VLM policy $\pi_\theta$ outputting $\boldsymbol{v}^{\text{out}}$ with input image $o$ and prompt $\boldsymbol{v}^{\text{in}}$.

## 4 Training VLMs with RL

Compared to classic MLP-based policy networks [53–55, 19], a natural advantage of VLM policies is that they can leverage CoT reasoning for efficient exploration, by performing intermediate reasoning steps that lead to the final decision. However, training a VLM policy $\pi_\theta$ with RL presents additional challenges. First, the VLM policy $\pi_\theta(o, \boldsymbol{v}^{\text{in}})$ directly generates open-ended text rather than vectorized actions in classic policy gradient-based RL methods [53–55, 19], complicating direct interaction with the environment. Even with a parsing mechanism $f : \mathcal{V}^n \to \mathcal{A}$ that maps open-ended text $\boldsymbol{v}^{\text{out}}$ to a *legal* action $a$ for interaction with the environment, it remains unclear how to estimate the action probability $\pi_\theta(a|o, \boldsymbol{v}^{\text{in}})$ from the text generation process.

Figure 2 presents an overview of our framework, leveraging the CoT reasoning and addressing the two aforementioned challenges. We design a task-specific prompt $\boldsymbol{v}^{\text{in}}$ that requires the VLM to generate a formatted output $\boldsymbol{v}^{\text{out}}$, including the CoT reasoning. Next, we adopt a post-processing function $f$ to parse open-ended text into a *legal* action $a_t$ that can directly interact with the environment. To compute $\pi_\theta(a|o, \boldsymbol{v}^{\text{in}})$, we develop a method to estimate its value based on the probability of each output token in $\boldsymbol{v}^{\text{out}}$.

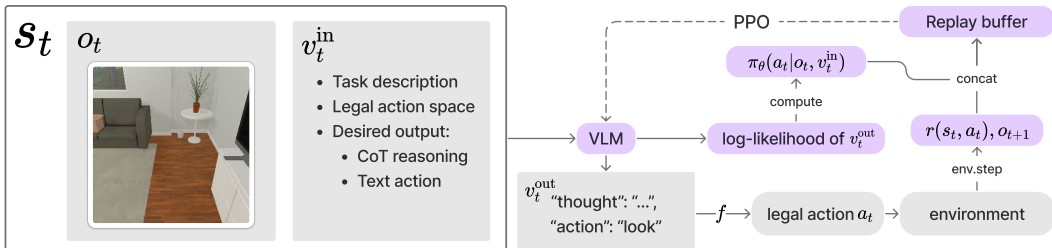

Figure 2: **A diagram of the proposed RL fine-tuning framework.** At time step $t$, the state $s_t$ contains an input prompt $\boldsymbol{v}_t^{\text{in}}$ and a visual observation $o_t$. The VLM takes $s_t = [o_t, \boldsymbol{v}_t^{\text{in}}]$ as input and outputs open-ended text $\boldsymbol{v}_t^{\text{out}}$ containing the CoT reasoning, keywords `"action"` : `"$a_t$"`, and the log-likelihood of $\boldsymbol{v}_t^{\text{out}}$. We first apply a post-processing function $f$ on $\boldsymbol{v}_t^{\text{out}}$, to obtain a *legal* action $a_t$ which can interact with the environment. Then, we input $a_t$ to the environment for obtaining reward $r(s_t, a_t)$ and the next observation $o_{t+1}$. Afterward, we devise a method to compute a numerical value of $\pi_\theta(a_t|o_t, \boldsymbol{v}_t^{\text{in}})$. Finally, we use $r(s_t, a_t)$ and $\pi_\theta(a_t|o_t, \boldsymbol{v}_t^{\text{in}})$ for the RL training.

The remaining Section is structured as follows. First, we describe the format of our input prompt $\boldsymbol{v}_t^{\text{in}}$ and the desired output $\boldsymbol{v}_t^{\text{out}}$ (Section 4.1). Next, we present the post-processing function $f$ (Section 4.2). Then, we introduce a method to compute a numerical value of $\pi_\theta(a_t|o_t, \boldsymbol{v}_t^{\text{in}})$ (Section 4.3). Finally, we conclude our framework in Algorithm 1 (Section 4.4).

### 4.1 Prompt Design for Domain-Specific Outputs

For each task $\mathcal{M}$, our input prompt $\boldsymbol{v}_t^{\text{in}}$ contains a description of the task, the legal action space of the current observation, and the desired output format (including the CoT reasoning). Our desired output $\boldsymbol{v}_t^{\text{out}}$, contains a CoT reasoning followed by the keywords `"action"` : `"$a_t$"` for post-processing. Figure 3 provides an example of our input prompt $\boldsymbol{v}_t^{\text{in}}$ and the desired formatted output $\boldsymbol{v}_t^{\text{out}}$. In particular, we define a function $h$ which constructs $\boldsymbol{v}_t^{\text{in}}$ from the current observation $o_t$: $\boldsymbol{v}_t^{\text{in}} = h(o_t)$, to accommodate for tasks that may contain observation-dependent information.[3] We provide additional examples of $\boldsymbol{v}^{\text{in}}$ and $\boldsymbol{v}^{\text{out}}$ in Appendix B.

---

[3]E.g., the `alfworld` environment (to be introduced in Section 5.2) contains an observation-dependent *admissible action* space.

Figure 3: **A template of our input prompt and output text.** The blue part represents the CoT reasoning and the red part is the text-based action. Note that the CoT reasoning may contain **other task-specific descriptions**, see Appendix B for more details.

## 4.2 Post-Processing Open-Ended Text for Legal Actions

Our post-processing mechanism involves both $v_t^{\text{in}}$ and $f$. In the input prompt $v_t^{\text{in}}$, we directly ask the VLM to output a text-based action in the format of "`action`" : "$a_t$" (see Figure 1 and Figure 2 for examples). After obtaining $v_t^{\text{out}}$, our post-processing function $f$ directly searches for the text-based keywords "`action`" : "$a_t$" from $v_t^{\text{out}}$, and maps it to a legal action $a_t$, either in symbolic or in text depending on the task of interest. For the case shown in Figure 1, $f$ will map $v_t^{\text{out}}$ to the symbolic operator that represents the action "`stand`" in the `Blackjack` task (to be introduced in Section 5.1), as the `Blackjack` task takes symbolic actions as input. For the `alfworld` [58] environment shown in Figure 2, $f$ will map $v_t^{\text{out}}$ to the text "`look`", because the `alfworld` environment takes text-based actions as inputs.

However, VLMs are not always guaranteed to generate a $v_t^{\text{out}}$ that contains the keywords "`action`" : "$a_t$", even when we explicitly request a formatted output from $v_t^{\text{in}}$. To continue the RL training when $v_t^{\text{out}}$ does not contain any legal action, we perform *random exploration* by selecting a legal action $a_t \in \mathcal{A}$ uniformly at random. Mathematically, $f$ is defined as follows:

$$f(\boldsymbol{v}^{\text{out}}) = \begin{cases} a, & \text{if "action" : "a"} \in \boldsymbol{v}^{\text{out}}, \\ \texttt{Unif}(\mathcal{A}), & \text{otherwise.} \end{cases} \tag{4.1}$$

## 4.3 Estimating Action Probabilities of VLM Policies

To estimate the action probability $\log \pi_\theta(a_t|o_t, \boldsymbol{v}_t^{\text{in}})$ (or equivalently $\log \pi_\theta(a_t|o_t, \boldsymbol{v}_t^{\text{in}})$) for policy gradient-based methods [55], a naïve calculation is directly using $\log \pi_\theta(\boldsymbol{v}_t^{\text{out}}|o_t, \boldsymbol{v}_t^{\text{in}})$ as $\log \pi_\theta(a_t|o_t, \boldsymbol{v}_t^{\text{in}})$, by summing the log-likelihood of all tokens in $\boldsymbol{v}_t^{\text{out}}$. This is because

$$\log \pi_\theta(\boldsymbol{v}_t^{\text{out}}|o_t, \boldsymbol{v}_t^{\text{in}}) = \log \frac{P(o_t, \boldsymbol{v}_t^{\text{in}}, \boldsymbol{v}_t^{\text{out}})}{P(o_t, \boldsymbol{v}_t^{\text{in}})}$$

$$= \log \left[ \frac{P(o_t, \boldsymbol{v}_t^{\text{in}}, \boldsymbol{v}_{[:n]})}{P(o_t, \boldsymbol{v}_t^{\text{in}}, \boldsymbol{v}_{[:n-1]})} \cdots \frac{P(o_t, \boldsymbol{v}_t^{\text{in}}, \boldsymbol{v}_{[:2]})}{P(o_t, \boldsymbol{v}_t^{\text{in}}, \boldsymbol{v}_{[:1]})} \frac{P(o_t, \boldsymbol{v}_t^{\text{in}}, \boldsymbol{v}_{[:1]})}{P(o_t, \boldsymbol{v}_t^{\text{in}})} \right] = \sum_{i=1}^{n} \log \left[ \frac{P(o_t, \boldsymbol{v}_t^{\text{in}}, \boldsymbol{v}_{[:i]})}{P(o_t, \boldsymbol{v}_t^{\text{in}}, \boldsymbol{v}_{[:i-1]})} \right]. \tag{4.2}$$

In the equation above, we use $\boldsymbol{v}$ to denote the output token $\boldsymbol{v}_t^{\text{out}}$ for simplicity, and we use $\boldsymbol{v}_{[:i]}$ to denote the first $i$ tokens in $\boldsymbol{v}_t^{\text{out}}$, and we slightly abuse our notion by using $P(o_t, \boldsymbol{v}_t^{\text{in}}, \boldsymbol{v}_{[:0]})$ to denote $P(o_t, \boldsymbol{v}_t^{\text{in}})$ in the log summation. Hence, a natural way to compute a numerical value for $\log \pi_\theta(a_t|o_t, \boldsymbol{v}_t^{\text{in}})$ is $\sum_{i=1}^{n} \log \left[ \frac{P(o_t, \boldsymbol{v}_t^{\text{in}}, \boldsymbol{v}_{[:i]})}{P(o_t, \boldsymbol{v}_t^{\text{in}}, \boldsymbol{v}_{[:i-1]})} \right]$.

However, the naïve calculation $\log \pi_\theta(a_t|o_t, \boldsymbol{v}_t^{\text{in}}) \leftarrow \sum_{i=1}^{n} \log \left[ \frac{P(o_t, \boldsymbol{v}_t^{\text{in}}, \boldsymbol{v}_{[:i]})}{P(o_t, \boldsymbol{v}_t^{\text{in}}, \boldsymbol{v}_{[:i-1]})} \right]$ may not be ideal for computing $\pi_\theta(a_t|o_t, \boldsymbol{v}_t^{\text{in}})$ since our formatted output $\boldsymbol{v}_t^{\text{out}}$ also contains CoT reasoning. This is because in $\boldsymbol{v}_t^{\text{out}} = [v_t^{\text{tht}}, v_t^{\text{act}}]$, the CoT reasoning tokens $v_t^{\text{tht}}$ are generally much

longer than the action tokens $\boldsymbol{v}_t^{\mathrm{act}}$ (see the blue and red parts in Figure 3 for examples, and see Table 1 for a relative scaling of their sum log-likelihood). Hence the naïve computation $\log \pi_\theta(a_t|o_t, \boldsymbol{v}_t^{\mathrm{in}}) \leftarrow \log \pi_\theta(\boldsymbol{v}_t^{\mathrm{tht}}|o_t, \boldsymbol{v}_t^{\mathrm{in}}) + \log \pi_\theta(\boldsymbol{v}_t^{\mathrm{act}}|o_t, \boldsymbol{v}_t^{\mathrm{in}}, \boldsymbol{v}_t^{\mathrm{tht}})$ will make $\log \pi_\theta(a_t|o_t, \boldsymbol{v}_t^{\mathrm{in}})$ largely determined by the CoT tokens $\log \pi_\theta(\boldsymbol{v}_t^{\mathrm{tht}}|o_t, \boldsymbol{v}_t^{\mathrm{in}})$, which is practically undesirable because our post-processing function $f$ only relies on $\boldsymbol{v}_t^{\mathrm{act}}$ for decision-making.

As shown in Table 1, $\log \pi_\theta(\boldsymbol{v}_t^{\mathrm{tht}}|o_t, \boldsymbol{v}_t^{\mathrm{in}})$ typically has a much larger magnitude than $\log P(\boldsymbol{v}_t^{\mathrm{act}}|o_t, \boldsymbol{v}_t^{\mathrm{in}}, \boldsymbol{v}_t^{\mathrm{tht}})$ across all tasks we have tested (in terms of absolute value). Hence, to mitigate the effect of the CoT tokens, we adopt a scaling factor $\lambda \in [0, 1]$ to scale down $\log \pi_\theta(\boldsymbol{v}_t^{\mathrm{tht}}|o_t, \boldsymbol{v}_t^{\mathrm{in}})$ for obtaining a regularized version of $\log \pi_\theta(a_t|o_t, \boldsymbol{v}_t^{\mathrm{in}})$, which results in

| log | NL | BJ | EZP | P24 | ALF |
|-----|-----|-----|-----|-----|-----|
| $\boldsymbol{v}_t^{\mathrm{tht}}$ | -3.4 | -2.2 | -9.0 | -37.6 | -20.3 |
| $\boldsymbol{v}_t^{\mathrm{act}}$ | 0.0 | 0.0 | 0.0 | 0.0 | -0.4 |

Table 1: **The absolute values of sum log probability of $\boldsymbol{v}_t^{\mathrm{tht}}$ is much larger than $\boldsymbol{v}_t^{\mathrm{act}}$.** Each number is averaged among 1000 samples on our evaluation tasks to be introduced in Section 5.

$$
\begin{aligned}
&\log \pi_\theta(a_t|o_t, \boldsymbol{v}_t^{\mathrm{in}}) \\
&\leftarrow \lambda \log \pi_\theta(\boldsymbol{v}_t^{\mathrm{tht}}|o_t, \boldsymbol{v}_t^{\mathrm{in}}) + \log \pi_\theta(\boldsymbol{v}_t^{\mathrm{act}}|o_t, \boldsymbol{v}_t^{\mathrm{in}}, \boldsymbol{v}_t^{\mathrm{tht}}).
\end{aligned}
\tag{4.3}
$$

Empirically, we observe the scaling factor $\lambda$ could largely affect the final performance. As we will show in Section 6.2, choosing an extreme $\lambda$ value (close to 1 or 0) will degrade overall performance. All of our experiments adopt $\lambda \in [0.2, 0.5]$.

### 4.4 Formal Implementation

Putting the prompt construction function $h$ (Section 4.1), the post-processing function $f$ (Section 4.2), and the computation of $\pi_\theta(a_t|o_t, \boldsymbol{v}_t^{\mathrm{in}})$ (Section 4.3) together, we conclude our method in Algorithm 1.

---

**Algorithm 1** Training VLM with RL

---

1: **Input:** An environment `env`, an initial VLM with parameters $\theta_0$.
2: **Input:** A post-processing function $f$, a CoT reasoning scaling factor $\lambda$.
3: **Input:** Replay buffer size $B$, maximum episode length $T$.
4: **for** $k = 0, \dots, K - 1$ **do**
5:      $t = 0$             ▷ Reset RL time step
6:      $o_t = \texttt{env.reset}()$             ▷ Reset the initial state
7:      $\boldsymbol{v}_t^{\mathrm{in}} = h(o_t)$             ▷ Generate $\boldsymbol{v}_t^{\mathrm{in}}$ from $o_t$, $h$ is defined in Section 4.1
8:      $\mathcal{B}_k = \emptyset$             ▷ Initialize an on-policy replay buffer
9:      **while** $|\mathcal{B}_k| \leq B$ **do**
10:          $\boldsymbol{v}_t^{\mathrm{out}} = \pi_{\theta_k}(o_t, \boldsymbol{v}_t^{\mathrm{in}})$             ▷ Generate text output
11:          $a_t = f(\boldsymbol{v}_t^{\mathrm{out}})$          ▷ Obtain a legal action from $\boldsymbol{v}_t^{\mathrm{out}}$, $f$ is defined in Equation 4.1
12:          $\log \pi_{\theta_k}(a_t|o_t, \boldsymbol{v}_t^{\mathrm{in}}) = \lambda \log \pi_{\theta_k}(\boldsymbol{v}_t^{\mathrm{tht}}|\boldsymbol{v}_t^{\mathrm{in}}) + \log \pi_{\theta_k}(\boldsymbol{v}_t^{\mathrm{act}}|o_t, \boldsymbol{v}_t^{\mathrm{in}}, \boldsymbol{v}_t^{\mathrm{tht}})$    ▷ Equation 4.2
13:          $r_t, o_{t+1} = \texttt{env.step}(a_t)$
14:          $\mathcal{B}_k = \mathcal{B}_k \cup \{(o_t, a_t, r_t, \boldsymbol{v}_t^{\mathrm{out}}, \log \pi_{\theta_k}(a_t|o_t, \boldsymbol{v}_t^{\mathrm{in}}))\}$          ▷ Add data to the buffer $\mathcal{B}_k$
15:          $t = t + 1$
16:          **if** $t = T$ **then**
17:              $t = 0$             ▷ Reset RL time step if the maximum step is reached
18:              $o_0 = \texttt{env.reset}()$             ▷ Reset environment
19:          **end if**
20:          $\boldsymbol{v}_t^{\mathrm{in}} = h(o_t)$             ▷ Prepare the next $\boldsymbol{v}_t^{\mathrm{in}}$
21:      **end while**
22:      Run PPO [55] with data $\mathcal{B}_k$ to obtain $\theta_{k+1}$
23: **end for**
24: **Output:** $\theta_K$.

---

## 5 Evaluation Tasks

How does our method improve a VLM's decision-making capabilities in tasks that require fine-grained vision-language reasoning or semantic understanding? To study this question, we adopt two different domains: `gym_cards` and `alfworld` [58]. Our original `gym_cards` domain is a "gym-like" environment [8] containing four tasks designed to test the decision-making capabilities of VLMs.

These tasks require fine-grained visual-language reasoning, specifically focusing on recognizing numbers for arithmetic reasoning. In addition, we also adopt alfworld [58], which assesses the decision-making capabilities of VLMs in an embodied AI setting that demands visual semantic understanding. We present some examples of the visual observations of each task in Figure 4. We do not include standard image-based Atari benchmarks [5, 39] due to limited computation resources.[4]

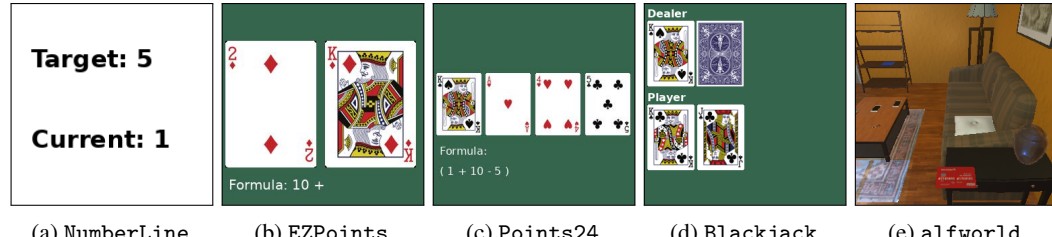

| (a) NumberLine | (b) EZPoints | (c) Points24 | (d) Blackjack | (e) alfworld |

Figure 4: **Examples of observation of our evaluation tasks**. (a)-(d) are from our original gym_cards domain. (a)-(c) are deterministic tasks with *increasing difficulties*; (d) is a stochastic task.

## 5.1 Gym Cards

Our gym_cards domain is designed to evaluate a VLM's decision-making capabilities requiring fine-grained vision recognition and language reasoning. More precisely, tasks in the gym_cards domain require the VLM to recognize the numbers (potentially from cards) and utilize the numbers for language reasoning. As depicted in Figure 4, the first three tasks—NumberLine, EZPoints, and Points24—are deterministic, and developed to assess the VLMs' ability to identify and process numbers or mathematical operators at each time step. These tasks increase in complexity: NumberLine requires recognition of two numbers in an image, EZPoints involves identifying numbers from two cards, and Points24 extends to recognizing four cards. The Blackjack task challenges the VLM further by requiring the agent to reason based on visual information and adapt to stochastic outcomes. This subsection outlines the goals of each task, and we leave the detailed descriptions of their state spaces, action spaces, and reward functions to Appendix B.1.

**NumberLine.** In this task, **the goal is to move a number to the target on a synthetic number line.** At each state $s_t$, the visual observation $o_t$ contains two lines of text: "Target: $x$" and "Current: $y_t$". The agent needs to move the current number $y_t$ to the target number $x$, by outputting text $\boldsymbol{v}_t^{\text{out}}$ that interacts with the discrete action space $\{$"+", "−"$\}$. Mapping the $\boldsymbol{v}_t^{\text{out}}$ to "+" or "−" will increase or decrease the current number by 1, respectively.

**EZPoints.** In this task, **the goal is to output a formula using the numbers in the cards that evaluates to 12.** At each state $s_t$, the agent observes an image of two cards and a text version of (potentially incomplete) "formula" below the cards. The goal is to use *all* numbers in the cards (only once) to compute 12. The action space contains natural numbers in $[1, 10]$, as well as operator in $\{$"+", "∗", "="$\}$. At each state $s_t$, only operators and numbers that appear in the cards are *legal* actions, and "J", "Q", or "K" are treated as "10". In particular, if the output text $\boldsymbol{v}_t^{\text{out}}$ is mapped to a legal action $a_t$ at state $s_t$, the text version of $a_t$ will be appended to the "formula" in the current image of $s_t$ resulting $s_{t+1}$, otherwise $s_{t+1}$ will remain the same as $s_t$.

**Points24.** In this task, **the goal is to output a formula using the numbers in the cards that evaluates to 24.** The Points24 task is a harder version of EZPoints as it contains 4 cards, hence requiring the VLMs to generate a longer formula. The rules of Points24 are similar to EZPoints, despite two minor differences: the Points24 task requires the VLM to compute a target number of 24, and its action space contains more operators: $\{$"+", "−", "∗", "/", "="$\}$.

**Blackjack.** In this task, **the goal is to win the current blackjack game.** At each state $s_t$, the visual observation $o_t$ consists of two cards (one face-down) from the dealer and all cards from the player.

---

[4]Image-based Atari tasks generally take at least 2 million environment steps to reach a reasonable performance [23]. Our method needs roughly 30 hours to run 15k environment steps due to the model size of the backbone VLMs, which requires roughly half a year to run 2 million environment steps.

The agent's goal in this task is to win the current game, by outputting text $v_t^{\text{out}}$ that can be mapped to {"stand", "hit"}. The agent will receive one more card if $v_t^{\text{out}}$ is mapped to "hit", and the game will terminate if $v_t^{\text{out}}$ is mapped to "stand".

## 5.2 ALFWorld

While the gym_cards domain is designed to assess the VLM's arithmetic reasoning requiring fine-grained visual recognition, the alfworld environment aims at testing VLM's decision-making tasks requiring visual semantic understanding.

**ALFWorld.** The ALFWorld embodied environment [58] is combines a text-based interactive environment [13] with a large vision-language instruction following dataset [57]. It contains 6 different types of goal-conditioned tasks ("Pick & Place", "Examine in Light", "Clean & Place", "Heat & Place", "Cool & Place", and "Pick Two & Place"), and **the agent's goal is to navigate in the environment via text-based actions** (e.g., "go to shelf 1", "examine sidetable 1"). Unlike our original gym_cards environment, where all states share the same action space, the alfworld environment contains a state-dependent *admissible action* space – some actions are only available at certain states. For example, if the agent's goal is to "put some pillows on armchair", then the agent can only put a pillow *after* picking up a pillow. Hence, to incorporate the state-dependent admissible action set, our prompt of alfworld asks the VLM to choose among an admissible action. See Figure 2 for an example of the visual observation of alfworld. We leave the detailed descriptions of the alfworld (state space, action space, reward functions, and the CoT prompt) to Appendix B.2.

# 6 Experimental Results

The first part of our experiment examines how our method improves the decision-making capabilities of VLMs (Section 6.1). The second part investigates the role of CoT reasoning in our method (Section 6.2). Details of our experimental setup are provided in Appendix C.

## 6.1 Improving VLM Decision-Making Capabilities

Does our method improve the decision-making capabilities of VLM agents across various domains? To investigate this, we assess how our method improves arithmetic tasks requiring fine-grained visual recognition in the gym_cards domain and visual semantic reasoning in the alfworld domain. The gym_cards experiments include deterministic tasks (NumberLine, EZPoints, and Points24, each with increasing difficulty) and a stochastic task (Blackjack). In the alfworld domain, we evaluate overall performance and detailed task-specific performance as discussed in Section 5.2. We instantiate our method on top of the llava-v1.6-mistral-7b [35] model and compare it against commercial models (GPT4-V and Gemini), a supervised fine-tuned version of the llava-v1.6-mistral-7b model (LLaVA-sft),[5] and a vanilla RL implementation using a CNN-based policy network (CNN+RL).[6] The final results and learning curves are presented in Table 2 and Figure 5. Details of the experimental setup are provided in Appendix C.

**Enhancing decision-making capabilities of VLM agents across various tasks.** As illustrated in Table 2 and Figure 5, our method demonstrates consistent improvement across various tasks, including deterministic (NumberLine and EZPoints)[7] or stochastic (Blackjack) arithmetic tasks and visual semantic reasoning task (alfworld). Specifically, our method improves the average performance from the initial LLaVA-sft model by **27.1%** on arithmetic tasks (18.4% → 45.5%) and

---

[5]To ensure the RL training starts from a model with reasonable instruction following capabilities [45], our RL training for VLM starts from the LLaVA-sft checkpoint of each task, we leave the detailed training pipeline of our method to Appendix C.1.

[6]The CNN-based method adopts the same CLIP vision encoder as LLaVA-7b. Additionally, for tasks that require text inputs (e.g., alfworld), we adopt the RoBERTa-base [36] model to encode the text feature and concatenate the text and CLIP visual features for downstream RL training. Details of our CNN-based model are provided to Appendix C.2.

[7]Although Points24 shares similar rules with EZPoints, it requires the VLM to recognize all four cards and generate much longer equations. Most failure cases in Points24 are caused by either inaccurate visual perception or flawed language reasoning. We provide some examples of these failures in Appendix C.5.

|  | gym_cards | | | | | alfworld | | | | | | | |
|---|---|---|---|---|---|---|---|---|---|---|---|---|---|
|  | NL | EZP | P24 | BJ | Avg. | Exp. Data | Pick | Look | Clean | Heat | Cool | Pick2 | Avg. |
| BUTLER$_g$ | - | - | - | - | - | ✓ | 33.0 | 17.0 | 26.0 | 70.0 | 76.0 | 12.0 | 22.0 |
| BUTLER | - | - | - | - | - | ✓ | 46.0 | 22.0 | 39.0 | 74.0 | 100.0 | 24.0 | 37.0 |
| CNN+RL | 87.1 | 0 | 0 | 38.8 | 31.5 | ✗ | 0 | 0 | 0 | 0 | 0 | 0 | 0 |
| GPT4-V | 65.5 | 10.5 | 0 | 25.5 | 25.4 | ✗ | 38.2 | 12.1 | **18.8** | 6.7 | 17.8 | 14.6 | 19.4 |
| Gemini | 82.5 | 2.0 | 0 | 30.0 | 28.6 | ✗ | 34.6 | **16.7** | 0 | 0 | 0 | 12.0 | 13.5 |
| LLaVA-sft | 24.8 | 23.0 | **2.6** | 23.1 | 18.4 | ✗ | 39.2 | 0 | 14.4 | 11.1 | 0 | **28.6** | 17.7 |
| Ours | **89.4** | **50.0** | 2.3 | **40.2** | **45.5** | ✗ | **47.4** | 14.7 | 10.4 | **14.4** | **18.8** | 18.0 | **21.7** |

Table 2: **Average episode success rates (%) of different methods on `gym_cards` and `alfworld`.** For all RL-based methods (CNN+RL and our method), we present the peak numbers (first 15k environment steps for the `gym_cards` and 5k environment steps for `alfworld`) from each training curve from Figure 5. We average the performance of all 4 tasks on `gym_cards` with equal weight. Due to the nature of the `alfworld` environment, where each subtask does not appear with equal probability, the average performance on `alfworld` is a weighted average among all types of tasks. We mark the BUTLER$_g$ and BUTLER agent [58] in gray since they require expert data, while the remaining methods do not require expert data. As discussed by Shridhar et al. [58], the performance discrepancy between BUTLER$_g$ and BUTLER happens due to different decoding strategies in evaluation strategies: BUTLER$_g$ uses greedy decoding, which may repeat failed actions, whereas BUTLER employs beam search during evaluation.

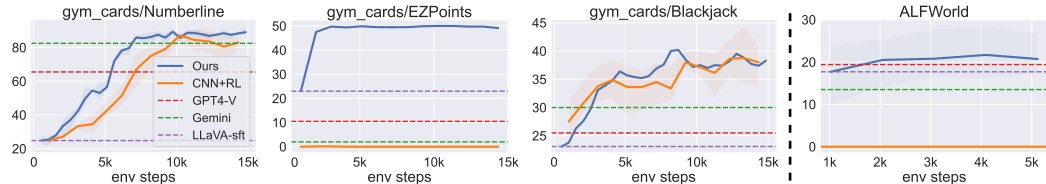

Figure 5: **Episode success rates (%) of different methods on `gym_cards` and `alfworld` during training.** Left to right: `gym_cards/Numberline`, `gym_cards/EZPoints`, `gym_cards/Blackjack`, and `alfworld` (all). The curves of `Points24` are not included because none of the tested methods achieve reasonable performance.

**4.0%** on visual semantic decision-making task (17.7% → 21.7%). In addition, our method also achieves the best performance among all comparative methods, surpassing the second-best method by 14.0 (CNN+RL) on `gym_cards` and 2.3% (GPT4-V) on `alfworld`.

## 6.2 Understanding the Role of the CoT Reasoning

In Section 6.1, we have demonstrated that our method improves the arithmetic and visual semantic reasoning capabilities of VLM agents. Conceptually, our method can be viewed as an augmented version of the standard CNN-based RL, where the text output $[v^{\text{tht}}, v^{\text{act}}]$ (from Figure 3) serve as the text action $v^{\text{act}}$, augmented by CoT reasoning $v^{\text{tht}}$. This raises an important question: How does the CoT reasoning $v^{\text{tht}}$ influence the overall performance of our method? To assess the impact of CoT reasoning on our method's performance, we conduct two sets of ablation experiments. The first set (presented in Table 3 and Figure 6) evaluates our method without the CoT reasoning, and the second part (shown in Figure 7) examines various scaling hyperparameters $\lambda$ for the log-likelihood of CoT tokens, as defined in Equation 4.3.

|  | gym_cards | | | | | alfworld | | | | | | |
|---|---|---|---|---|---|---|---|---|---|---|---|---|
| CoT | NL | EZP | P24 | BJ | Avg. | Pick | Examine | Clean | Heat | Cool | Pick 2 | Avg. |
| ✓ | **89.4** | **50.0** | **2.3** | 40.2 | **45.5** | **47.4** | **14.7** | **10.4** | **14.4** | **18.8** | **18.0** | **21.7** |
| ✗ | 26.9 | 29.9 | 0 | **40.4** | 24.3 | 40.5 | 12.0 | 2.8 | 8.5 | 14.4 | 17.7 | 16.3 |
| Diff. (✓ - ✗) | +62.5 | +20.1 | +2.3 | -0.2 | +21.2 | +6.9 | +2.7 | +7.6 | +5.9 | +4.4 | +0.3 | +5.4 |

Table 3: **Episode success rates (%) of our method with and without CoT reasoning.** We report the *best* results from Figure 6 (first 15k environment steps for the `gym_cards` and 5k environment steps for `alfworld`).

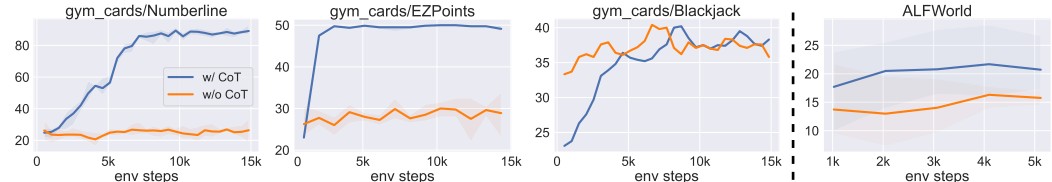

Figure 6: **Training curves of our method without and without the CoT reasoning.** Left to right: `gym_cards/Numberline`, `gym_cards/EZPoints`, `gym_cards/Blackjack`, and `alfworld` (all). The curves of `Points24` are not included because none of the tested methods achieve reasonable performance.

**The crucial role of CoT reasoning in performance improvement.** As presented in Table 3 and Figure 6, the performance of our method significantly decreases without the CoT reasoning.[8] Besides the improvement in the final performance, CoT reasoning appears to be a crucial component for deterministic arithmetic tasks (`NumberLine` and `EZPoints`), as our method fails to improve these two tasks without the CoT reasoning.

**The importance of moderate scaling factors $\lambda$.** As discussed in Section 4.3, integrating CoT reasoning into our framework involves tuning an additional hyperparameter, $\lambda \in [0, 1]$ (proposed in Equation 4.3). To identify an optimal range for $\lambda$, we conduct experiments assessing the impact of various $\lambda$. Our results in Figure 7 indicate that a moderate $\lambda$ (between 0.3 and 0.5) enables effective training on the `NumberLine` task. Conversely, our method fails when $\lambda$ is set too large ($\geq 0.7$) or too small ($\leq 0.1$), and we empirically find that an optimal $\lambda$ typically falls within 0.2 to 0.5. This is because a large $\lambda$ results in an estimate of $\log \pi_\theta(a_t|o_t, \boldsymbol{v}_t^{\text{in}})$ being overly influenced by $\log \pi_\theta(\boldsymbol{v}_t^{\text{tht}}|o_t, \boldsymbol{v}_t^{\text{in}})$, while a small $\lambda$ value causes $\pi_\theta$ to be predominantly affected by $\log \pi_\theta(\boldsymbol{v}_t^{\text{act}}|o_t, \boldsymbol{v}_t^{\text{in}}, \boldsymbol{v}_t^{\text{tht}})$, thereby reducing the effect of the CoT reasoning in RL training.

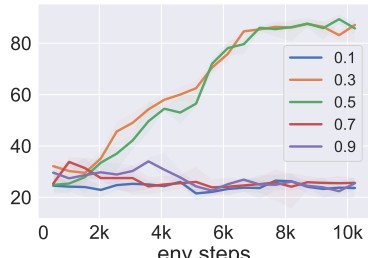

Figure 7: **Episode success rates (%) of our method under different $\lambda$ on `NumberLine`.**

# 7 Conclusions, Limitations, and Future Directions

In this paper, we introduce an algorithmic framework that directly fine-tunes VLMs using RL, with the help of the VLM's CoT reasoning capability. Empirical results demonstrate that our method can enhance the decision-making abilities of VLMs across diverse domains that require fine-grained visual recognition or visual semantic understanding. In addition, we demonstrate that CoT reasoning is a crucial component for enabling RL training, allowing 7b VLMs to outperform established commercial models such as GPT-4V and Gemini on most tasks. While our results suggest that CoT reasoning is crucial to the performance improvement of VLM training with RL, we have not extensively explored the effects of different prompting techniques in this work, which will be an interesting future direction. The performance gain of our method is also limited by the size of the action space and the difficulties of the task. For example `alfworld` does not enjoy as much performance gain as `gym_cards`, since `alfworld` is a multi-task environment and it has a much larger action space than `gym_cards`.

# 8 Acknowledgement

We would like to thank William Chen, Kuan Fang, Aviral Kumar, Qiyang Li, Fangchen Liu, Oier Mees, Seohong Park, Karl Pertsch, Haozhi Qi, Chun-Hsiao Yeh, and Andrea Zanette for the early discussions and suggestions on the project. A.S. is partly supported by AI2 Young Investigator Grant, and a Gemma Academic Program Award. S.X. is partly supported by an Amazon research award and the Google TRC program. This research was supported by NSF RI IIS-2246811, AFOSR FA9550-22-1-0273, the joint Simons Foundation-NSF DMS grant #2031899, the ONR grant N00014-22-1-2102, Tsinghua Berkeley Shenzhen Institute (TBSI) Research Fund, and the Hong Kong Center for Construction Robotics Limited (HKCRC) Award 052245. We would also like to thank Hyperbolic Labs for the computing support.

---

[8]Except for the `Blackjack` task, where the peak performance without CoT is slightly better (+0.2%).

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

# A Contributions

- **YXZ:** proposed, led, and managed the project; integrated all code bases; ran all ablations for method development; babysat all experiments; implemented the post-processing function $f$; proposed and implemented the scaling factor $\lambda$ for action tokens; beautified the gym_cards environment; maintained all codebases; wrote the major part of the paper.

- **HB:** set up the infrastructure and initial experiments for supervised fine-tuning before RL training; maintained all codebases; partially wrote the paper.

- **ZL:** set up the alfworld environment; set up major infrastructures for data collection; maintained all codebases; partially wrote the paper.

- **JP:** proposed the CoT idea for end-to-end RL training; optimized the RL training framework with quantization and enabled distributed training; implemented the initial version of the gym_cards environment; partially wrote the paper.

- **ST:** maintained the usage of LLaVA repo [34, 33, 35]; implemented the queries for GPT4-V and Gemini; partially wrote the paper.

- **YFZ:** implemented the initial version of RL training on LLaVA; partially wrote the paper.

- **AS, SX, YL, YM, SL:** provided suggestions for the project. **AS, SX, SL** also provided feedbacks on writing. **YM, SL** inspired **YXZ** to initiate the entire project.

# B Additional Details of the Evaluation Tasks

## B.1 Gym Cards

### B.1.1 NumberLine

**State and action space.** In the NumberLine task, the visual observation at each state $s_t$ contains two lines of text: "Target: $x$" and "Current: $y_t$", where $x, y_t$ are both integers such that $x, y_t \in [0, n_{\max}]$, where $n_{\max}$ is an environment input variable that controls the maximum position of the numbers. The goal is to move the current number $y_t$ to the target number $x$, by sequentially choosing actions from the discrete action space $\{$"+", "$-$"$\}$. We set $n_{\max} = 5$ for all experiments in this work, but $n_{\max}$ can be set to any positive integers. Choosing "+" or "$-$" will increase or decrease the current number $y_t$ by 1, respectively, and the agent will stay at the boundary if it takes an action that attempts to cross the boundary (e.g., taking $a_t =$ "+" when $y_t = n_{\max}$ or $a_t =$ "$-$" when $x_t = 0$). See an example of the state action transition in Figure 8.

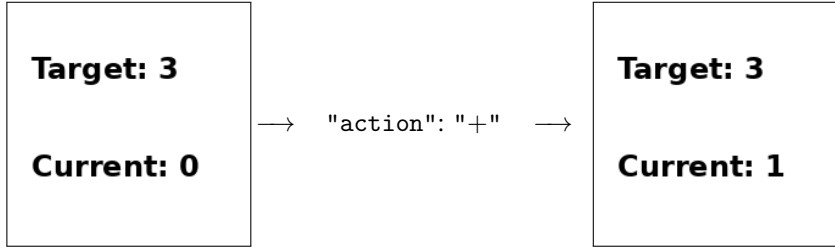

Figure 8: **An example of the transition in NumberLine.**

**Reward functions and the CoT prompts.** An episode in NumberLine ends when the current number equals the target number or the maximum step $T = 2n_{\max}$ is reached. The agent receives a terminal reward of $r(s_t, a_t) = 1$ when $y_{t+1} = x$. The agent also receives a reward penalty of $r(s_t, a_t) = -1$ upon taking an incorrect action that does not result in a closer position to the target ($|x - y_t| \geq |x - y_{t+1}|$), otherwise the agent receives reward $r(s_t, a_t) = 0$. In the example provided above (Figure 8), the agent receives a reward $r = 0$, since it moves closer to the target, but not reaching the target yet. For the NumberLine task, we adopt the following CoT prompt in Figure 9, and for the case without CoT reasoning (discussed in Section 6.2), we use the same prompt but without the blue CoT reasoning parts.

---

**CoT prompt $v_t^{\text{in}}$ for task** `NumberLine`

You are playing a game called number line. You will see a target number and a current number in the image. And your goal is to move the current number closer to the target by choosing either adding or subtracting one to the current number. Your response should be a valid json file in the following format:

{
"current number": "x",
"target number": "x",
"thoughts": {first read out the current and target number, then think carefully about which action to choose},
"action": "-" or "+"
}

---

Figure 9: **Task-specific CoT prompt input $v_t^{\text{in}}$ for** `NumberLine`. The blue part represents the CoT reasoning and the red part is the text-based action.

### B.1.2 EZPoints

**State and action space.** In the `EZPoints` task, the agent will observe an image of two cards and a text version of "formula" below the cards, at each state $s_t$. The goal is to use the cards in the image to compute a target number of 12 and we view {"J", "Q", "K"} as "10". The action space of `EZPoints` is {"1", "2", ..., "10", "+", "*", "="} and each number in the cards can *only be used once*. Any action attempting to either select a number not shown in the cards or use a card more than once are *illegal*. At $s_t$, if a *legal* action $a_t$ is taken, the action will be appended to the text "formula" in $s_t$ and becomes the next state $s_{t+1}$. On the other hand, when an illegal action is taken, $s_{t+1}$ will remain the same as $s_t$. All images generated from the `EZPoints` environment are guaranteed to have a viable solution for computing 12.

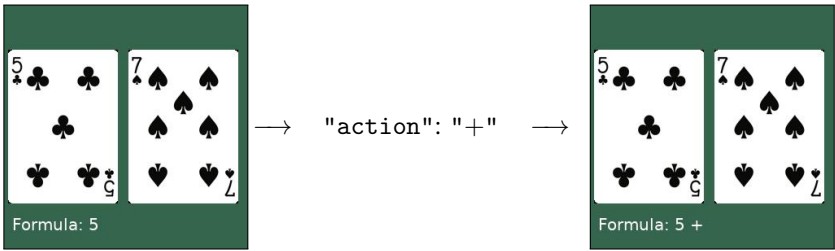

Figure 10: **An example of the transition in** `EZPoints`.

**Reward functions and the CoT prompts.** An episode terminates when "=" is taken or the maximum step $T = 5$ is reached. The agent receives a reward of $r = -1$ upon taking an *illegal* action, and $r = 0$ while taking a legal action. When "=" is taken, the agent will receive a positive reward $r = 10$ if the formula equals 12, and $r = -1$ otherwise. For the `EZPoints` task, we adopt the following CoT prompt in Figure 11, and for the case without CoT reasoning (discussed in Section 6.2), we use the same prompt but without the blue CoT reasoning parts and the brown part in Figure 11 is the text version of the current formula directly extracted from the current state $s_t$.

Figure 11: **Task-specific CoT prompt input $v_t^{in}$ for** `EZPoints` **given the observation in Figure 10.** The blue part represents the CoT reasoning, the red part is the text-based action, and the brown part is the state-dependent text from the formula in the image.

### B.1.3 Points24

**State and action space.** Similar to `EZPoints`, the goal of `Points24` is also to generate a formula to compute the target number of 24, using all four cards. `Points24` has a slightly larger action space: {"1", "2", ..., "10", "+", "−", "∗", "/", "(", ")", "="} and two more cards. Each number in the cards can *only be used once*. Similar to `EZPoints`, any action attempting to either select a number not shown in the cards or use a card more than once are *illegal*. At $s_t$, if a *legal* action $a_t$ is taken, the action will be appended to the text "formula" in $s_t$ and becomes the next state $s_{t+1}$. When an illegal action is taken, $s_{t+1}$ will remain the same as $s_t$. Different from `EZPoints` where all images are guaranteed to have a viable solution for computing 12, the images generated by `Points24` do not always have a viable solution to 24.

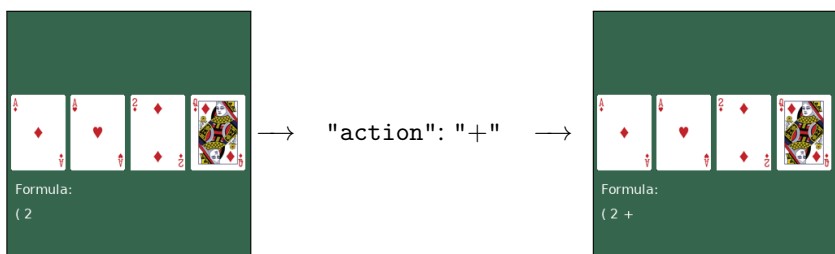

Figure 12: **An example of the transition in** `Points24`.

**Reward functions and the CoT prompts.** The reward functions and termination conditions of `Points24` are the same as those in `EZPoints`. An episode terminates when `"="` is taken or the maximum step $T = 20$ is reached. The agent receives a reward of $r = -1$ upon taking an *illegal* action, and $r = 0$ while taking legal actions. When `"="` is taken, the agent will receive a positive reward $r = 10$ when the formula equals 24, and $r = -1$ otherwise. For the `Points24` task, we adopt the following CoT prompt in Figure 13, and for the case without CoT reasoning (discussed in Section 6.2), we use the same prompt but without the blue CoT reasoning parts and the brown part in Figure 13 is the text version of the current formula directly extracted from the current state $s_t$. We also provide an additional feature that allows us to view {"J", "Q", "K"} as {"11", "12", "13"}, instead of {"10"}.



**CoT prompt $v_t^{\text{in}}$ for** `Points24`

You are an expert 24 points card game player. You are observing these four cards in the image. You are observing the current formula: '(2'. You can choose between ['1', '2', '3', '4', '5', '6', '7', '8', '9', '10', '+', '-', '*', '/', '(', ')', '=']. The number or operator you choose will be appended to the current formula. Note that 'J', 'Q', and 'K' count as '10'. Your goal is to output a formula that evaluates to 24, and each number can only be used once. Your response should be a valid json file in the following format:

{
"cards": [x, y, z, w],
"current formula": '(2'
"thoughts": {First check whether the current formula equals 24. If the current formula equals 24, output '='. Otherwise consider which number or operator should be appended to the current formula to make it equal 24.}
"action": "{number}" or "{operator}"
}



Figure 13: **Task-specific CoT prompt input $v_t^{\text{in}}$ for** `Points24` **given the observation in Figure 12.** The blue part represents the CoT reasoning and the red part is the text-based action, brown part is the state-dependent text that directly obtained from the formula in the image.

### B.1.4 Blackjack

**State and action space.** For the `Blackjack` task, the visual observation at state $s_t$ consists of two cards (one face-down) from the dealer and all cards from the player. The agent's goal in this task is to win the current game, by choosing actions in {"stand", "hit"}. The agent will receive a new card upon choosing "hit". See Figure 14 for an example transition.

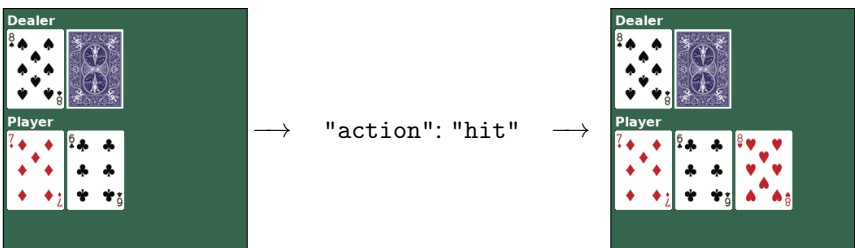

Figure 14: **An example of the transition in** `Blackjack`.

**Reward functions and the CoT prompts.** The game terminates when the player chooses "stand" or busts (total points exceed 21). We adopt the same reward function as the `Blackjack-v1` task in Gymnasiym [68], where $r(s_t, a_t) = 1, 0, -1$ upon win, draw, and loss, respectively. We also provide a similar feature as Gymnasium [68], where the "blackjack" winning (the agent win with an "A" and a "10", "J", "Q" or "K") reward $r$ of the player will become 1.5. In the example provided in Figure 14, the game has not terminated after taking the action "hit", hence the agent will not receive any rewards, even though it has total points of 21. For the `Blackjack` task, we adopt the following CoT prompt in Figure 15, and for the case without CoT reasoning (discussed in Section 6.2), we use the same prompt but without the blue CoT reasoning parts.



**CoT prompt $v_t^{\text{in}}$ for** `Blackjack`

You are a blackjack player. You are observing the current game state, you can choose between ['stand', 'hit']. Your response should be a valid json file in the following format:

{
"thoughts": "{first describe your total points and the dealer's total points then think about which action to choose}",
"action": "stand" or "hit"
}



Figure 15: **Task-specific CoT prompt input $v_t^{\text{in}}$ for** `Blackjack`. The blue part represents the CoT reasoning and the red part is the text-based action.

## B.2 ALFWorld

**State and action space.** Inherited from Text World [13], at each state $s_t$ of `alfworld`, the agent will observe an RGB image and text-based description. The action space of `alfworld` can be summarized these following format [58]: (1) `goto {recep}`; (2) `take {obj} from {recep}`; (3) `put {obj} in/on {recep}`; (4) `open {recep}`; (5) `close {recep}`; (6) `toggle {obj}{recep}`; (7) `clean {obj} with {recep}`; (8) `heat {obj} with {recep}`; (9) `cool {obj} with {recep}`, where `{obj}` and `{recep}` stands for objects and receptacles. See Figure 16 for an example of the state action transition in the `alfworld` environment.

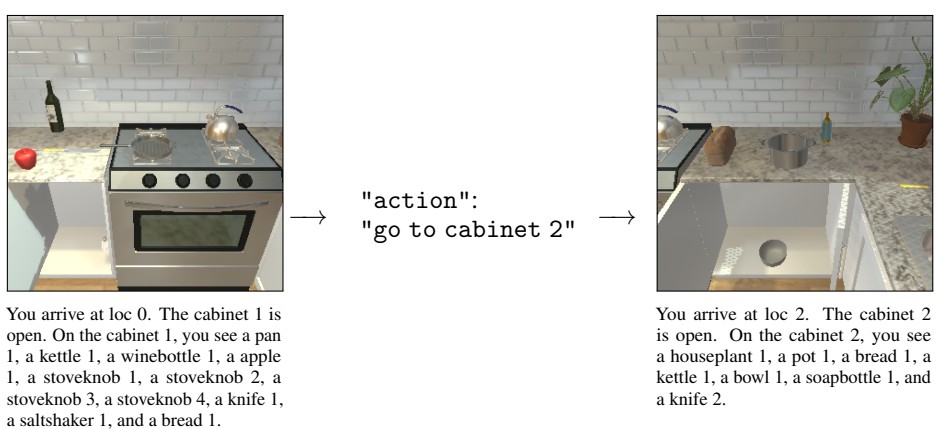

You arrive at loc 0. The cabinet 1 is open. On the cabinet 1, you see a pan 1, a kettle 1, a winebottle 1, a apple 1, a stoveknob 1, a stoveknob 2, a stoveknob 3, a stoveknob 4, a knife 1, a saltshaker 1, and a bread 1.

You arrive at loc 2. The cabinet 2 is open. On the cabinet 2, you see a houseplant 1, a pot 1, a bread 1, a kettle 1, a bowl 1, a soapbottle 1, and a knife 2.

Figure 16: **An example of the transition in `alfworld`.**

**Reward functions and the CoT prompts.** Each state $s \in \mathcal{S}$ of `alfworld` has a set of *admissible actions* $\mathcal{A}_{\text{adm}}(s)$, a final goal $g_{\text{task}}$, and subgoals $g_{\text{sub}}$. Since the goal of `alfworld` is to complete the language-based goal-conditioned tasks, we reward the agent upon reaching subgoals and completing the task, while penalizing the agent upon taking inadmissible actions. To summarize, we define the reward function of `alfworld` as $r(s_t, a_t, s_{t+1}|g_{\text{task}}) = 50 * \mathbf{1}\{s_{t+1} = g_{\text{task}}\} + \mathbf{1}\{s_{t+1} = g_{\text{sub}}\} - \mathbf{1}\{a_t \notin \mathcal{A}_{\text{adm}}(s_t)\}$. For the `alfworld` task, we adopt the following CoT prompt in Figure 17, and for the case without CoT reasoning (discussed in Section 6.2), we use the same prompt but without the blue CoT reasoning parts and the brown part in Figure 17 is the text description of the task directly extracted from the current state $s_t$.

Figure 17: **Task-specific CoT prompt input $v_t^{\text{in}}$ for alfworld given the observation in Figure 16.** The blue part represents the CoT reasoning and the red part is the text-based action, brown part is the state-dependent text that directly obtained from the text description and the admissible actions of the current state.

# C   Additional Details on the Experiments

We provide additional detailed of the experimental results in Section 6 here. Details of our experimental pipeline is provided in Section C.1, including preparing the initial SFT checkpoints and the RL training. Section C.2 contains details setup of all comparative methods. We list task-specific training details in Section C.3. We provide additional experimental results in Section C.4. Section C.5 lists several failure examples of the Points24 tasks.

## C.1   Experimental Pipeline

Our experiments adopt a similar pipeline as RLHF [45], where we first apply supervised fine-tuning (SFT) to the backbone llava-v1.6-mistral-7b model, before RL training. As outlined by Ouyang et al. [45], the RLHF training procedure consists of three distinct stages: SFT, learning reward models from human preference data, and applying RL with the learned reward models. Our pipeline is analogous to RLHF but without requiring the collection of human preference data for learning reward models, as we can directly collect rewards from the environment.[9] Consequently, our experimental pipeline only contains two stages: SFT and RL, which we will explain below.

**Supervised fine-tuning.** For the original gym_cards environment, we manually construct instruction-following data for all tasks following the format specified in Figure 3 of Section 4.1. As for alfworld, we use GPT4-V [44] to collect instruction following data for SFT. For all tasks, we prepare two versions of the instruction-following data, one with CoT and one without. We leave the details of the CoT prompts for each task, and the details of each fine-tuning dataset in Appendix D. After constructing the instruction-following data (with and without CoT), we fine-tune llava-v1.6-mistral-7b for 1 epoch on the collected data for each task and report the results for LLaVA-sft.

**RL training.** For each task, we start our RL training from the LLaVA-sft checkpoint. The LLaVA model [34] consists of three jointly trainable components, a CLIP vision encoder [49], an LLM

---

[9]We adopt the same pipeline for the evaluation without CoT reasoning (discussed in Section 6.2) while changing the data for SFT as well as $v^{\text{in}}$ (see more details on our SFT data and $v^{\text{in}}$ in Appendix D)

backbone [66, 67, 25], and an MLP projector that connects visual features and the word embeddings, and we directly apply PPO [55] to train all three components. Due to computation resource limitations, we instantiate our experiments via LoRA [21], with the LoRA configuration of $r = 128, \alpha = 256, \texttt{dropout} = 0.05$, for all trainable components. For the CoT coefficient $\lambda$, we set $\lambda = 0.5$ in the `gym_cards` domain and $\lambda = 0.2$ in `alfworld`.

## C.2 Experimental Setup for Comparative Methods

**GPT4-V and Gemini.** All of our experimental results on GPT4-V [44] and Gemini [18] are tested on March 15, 2024, using the same prompt for our RL training (see detailed prompts in Appendix D). For `gym_cards`, the numbers from both GPT4-V and Gemini are averaged among the same number of episodes: 200 episodes for deterministic tasks (`NumberLine`, `EZPoints` and `Points24`); 1000 episodes for stochastic task (`Blackjack`). As for `alfworld`, we report the performance of GPT4-V on all 1000 episodes we collected, see Appendix D.5 for our data collection on `alfworld` using GPT4-V. Due to the financial budget, we report the results of Gemini using 100 episodes.

**LLaVA-sft.** For each number of LLaVA-sft, we first collect the instruction-following dataset for each task and then fine-tune LLaVA-1.6-7b for 1 epoch on the collected data using the official LLaVA fine-tuning script.[10] Details of our data collection process is provided in Appendix D. We also *use the same LLaVA-sft checkpoint as initializations for the downstream RL training.*

**CNN-based RL.** Since the LLaVA-7b model adopts a CLIP ViT-L/14 vision encoder which is more powerful than vanilla CNN embeddings, we instantiate our CNN-based method using the feature from the same CLIP ViT-L/14 for a fair comparison. For tasks (`EZPoints`, `Points24`, and `alfworld`, see our detailed prompt in Appendix D) that require text inputs, we adopt the `RoBERTa-base` [36] model to encode the text feature and concatenate the text and CLIP visual features for downstream RL training. After obtaining the CLIP (potentially concatenated with text) features, we adopt 2 MLP layers followed by a fully connected layer to map the clip features into the action space. We adopt the PPO [55] implementation from Kostrikov [27] as the backbone RL algorithm. In addition, we adopt a `CosineAnnealingLR` learning rate scheduler, with the initial learning rate of $3e - 4$, the final learning rate of $1e - 8$, and the maximum learning rate step of 25. The remaining task specific hyperparameters are the same as the VLM case in Section C.3.

## C.3 General Setup for End-to-End RL Training

All experiments are conducted on an 8 A100s DGX machine (80G), while the maximum VRAM requirement is $< 40$G. Each curve from Figure 5 and 6 takes at most 36 hours to finish. We adopt DeepSpeed zero2 [51] for multi-gpu training. During our training for the VLM, we directly train all trainable components (vision encoder, LLM, and the MLP projector). We adopt an open-source implementation [27] for the PPO. Inspired by von Werra et al. [70], Castricato et al. [10], we apply a 3-layer MLP as the value head, on top of the output hidden states layer *before the output tokens*, to estimate the value function $V^{\pi_\theta}$. After obtaining the value estimate $V_\phi$, we adopt the generalized advantage estimator (GAE) [54] to estimate the return function $\hat{R}(s)$ and the advantage function $\hat{A}^{\pi_\theta}$ of $\pi_\theta$. In addition, we adopt a `CosineAnnealingLR` learning rate scheduler, with the initial learning rate of $1e - 5$, the final learning rate of $1e - 9$, and the maximum learning rate step of 25. For all experiments in the `gym_cards` and `alfworld` environment, we set the scaling hyperparameter $\lambda = 0.5, 0, 2$, respectively. The learning rate decay happens after every PPO update, which consists of 4 epochs of gradient updates with PPO. The number of data for on-policy training and batch size is task-dependent, we list them below.

**Numberline and Blackjack.** For `NumberLine` and `Blackjack`, our VLM training curves in Figure 5 use 4 GPUs. Our implementation naturally enables different random seeds on different GPUs, hence our VLM curves are averaged among 4 seeds. For one PPO update on each GPU, we collect 512 transitions, with a batch size of 128 per GPU (batch size = 512 in total). The episode return and success rate are averaged with `NumberLine`, `Blackjack` are averaged among 200 and 1000 episodes, respectively. We averaged the return of `Blackjack` on more episodes because

---

[10]`https://github.com/haotian-liu/LLaVA/blob/main/scripts/v1_5/finetune.sh`. We start from the llava-v1.6-mistral-7b instead of the v1.5 checkpoint in the script.

`Blackjack` contains stochastic while `NumberLine` is a deterministic task. We adopt the same number of transitions and batch size for the on-policy training in the CNN-based method on both tasks. The CNN-based methods are averaged among 4 random seeds as well.

**EZPoints and Points24.** For `EZPoints` and `Points24`, our VLM training curves in Figure 5 use 4 GPUs. Our implementation naturally enables different random seeds on different GPUs, hence our VLM curves are averaged among 4 seeds. For one PPO update on each GPU, we collect 1024 transitions, with a batch size of 128 per GPU (batch size = 512 in total). We use 1024 transitions because the episodes of `EZPoints` and `Points24` usually have longer horizons than `NumberLine` and `Blackjack`. The episode return and success rate are averaged with `EZPoints` and `Points24` are averaged among 200. We adopt the same number of transitions and batch size for the on-policy training in the CNN-based method on both tasks. The CNN-based methods are averaged among 4 random seeds as well.

**ALFWorld.** For the `alfworld` environment, each run of our VLM training curves in Figure 5 and Figure 19 are conducted on one GPU, and each curve is averaged among 4 seeds. We do not conduct multi-GPU training for `alfworld` because the on-policy sampling time has a huge variance on different GPUs, which will largely increase the synchronization time across different GPUs. For each PPO update, we collect 1024 transitions, and with a batch size of 256. The episode success rates are averaged among 200 episodes. We adopt the same number of transitions and batch size for the on-policy training in the CNN-based method on both tasks. The CNN-based methods are averaged among 4 random seeds as well.

## C.4 Additional Experimental Results

We provide some additional experimental results on the episode returns on the `gym_cards` and the task-specific training curves for `alfworld` here.

| | Episode Success Rate (%) | | | | Episode Return | | | |
|---|---|---|---|---|---|---|---|---|
| | NL | EZP | P24 | BJ | NL | EZP | P24 | BJ |
| CNN+RL | 87.1 | 0 | 0 | 38.8 | 0.79 | -1.02 | **-1.12** | -0.17 |
| GPT4-V | 65.5 | 10.5 | 0 | 25.5 | -0.59 | -1.30 | -4.39 | -0.44 |
| Gemini | 82.5 | 2.0 | 0 | 30.0 | 0.74 | -2.57 | -2.68 | -0.35 |
| LLaVA-sft | 24.8 | 23.0 | **2.6** | 23.1 | -2.30 | -0.50 | -13.52 | -0.50 |
| Ours | **89.4** | **50.0** | 2.3 | **40.2** | **0.87** | **4.46** | -11.84 | **-0.13** |

Table 4: **Average episode success rates and returns of different methods on `gym_cards`.** For all RL-based methods (CNN and our method), we report the *best* results in each training curve from Figure 18.

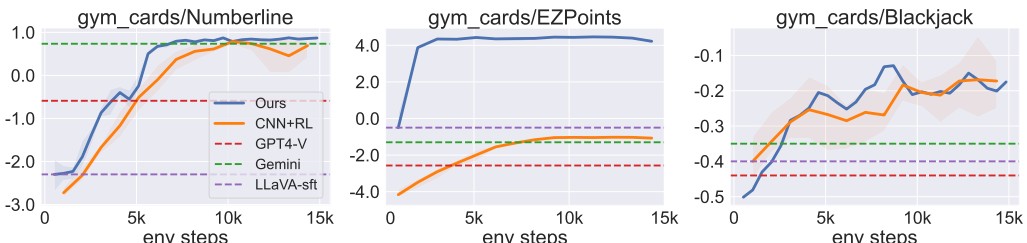

Figure 18: **Episode returns of different methods on `gym_cards`.** An extended version of Figure 5 containing episode success rates and returns.

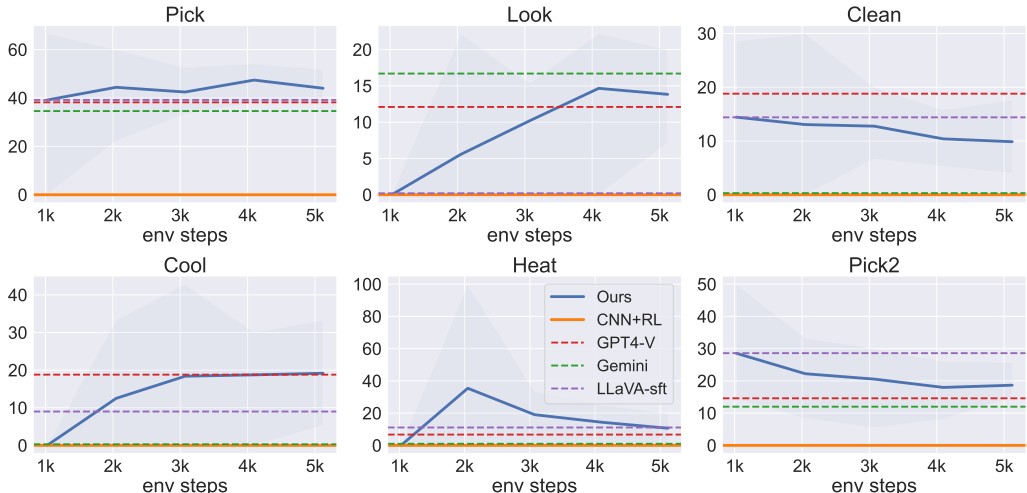

Figure 19: **Episode success rates our method on different types of tasks from `alfworld`.** The curves come from the same run as experiments from the `alfworld` figure of Figure 5.

## C.5 Failure Examples of `Points24`

---

**Ground Truth Cards: [2, 8, 5, J]**
**Error Type: Fail to recognize all numbers**

{
"cards": [10, 10, 5, 9],
"formula": "10+",
"thoughts": "'10+' is an incomplete formula, since '10+10-5+9=24', I should append '10' to the current formula",
"action": "10"
}

---

Figure 20: **Failure example of `Points24` due to incorrect perception.** In this example, the VLM fails to recognize the four numbers from the cards correctly as it only recognizes 2 numbers correctly. The reasoning part is correct, as it indeed generates a formula that equals 24.

---

**Ground Truth Cards: [J, K, 5, 6]**
**Error Type: Fail to recognize all numbers**

{
"cards": [10, 10, 5, 7],
"formula": "10+10+5+",
"thoughts": "'10+10+5' is an incomplete formula, since '10+10+5+7=24', I should append '7' to the current formula",
"action": "7"
}

---

Figure 21: **Failure example of `Points24` due to incorrect perception and reasoning.** In this example, the VLM fails to recognize the four numbers from the cards correctly as it only recognizes 3 numbers correctly. The reasoning part is also incorrect, 10+10+5+7 does not equal 24.

# D  Details on Supervised Fine-tuning Data

In this section, we provide detailed descriptions of the supervised fine-tuning data for all tasks.

## D.1  NumberLine

In the `NumberLine` task, we collect 20k instruction-following expert data samples using a task solver. See examples with and without CoT in Figure 22 and 23, respectively.

---

**Inputs:**
You are playing a game called number line. You will see a target number and a current number in the image. And your goal is to move the current number closer to the target by choosing either adding or subtracting one to the current number. Your response should be a valid json file in the following format:
{
"current number": "x",
"target number": "x",
"thoughts": {first read out the current and target number, then think carefully about which action to choose},
"action": "-" or "+"
}

**Target: 3**

**Current: 0**

---

**Responses:**
{
"current number": "0",
"target number": "3",
"thoughts": "The current number is 0, which is smaller than the target number is 3. To move the current number closer to the target, I should increase the current number by choosing +.",
"action": "+"
}

Figure 22: An example of the supervised fine-tuning data for `NumberLine` with CoT.

---

**Inputs:**
You are playing a game called number line. You will see a target number and a current number in the image. And your goal is to move the current number closer to the target by choosing either adding or subtracting one to the current number. Your response should be a valid json file in the following format:
{
"action": "-" or "+"
}

**Target: 3**

**Current: 0**

---

**Responses:**
{
"action": "+"
}

Figure 23: An example of the supervised fine-tuning data for `NumberLine` without CoT.

### D.2 Blackjack

In the `Blackjack` task, we collect 50k instruction-following expert data samples following an open-sourced solver.[11] See examples with and without CoT in Figure 24 and 25, respectively.

---

**Inputs:**

You are a blackjack player. You are observing the current game state, you can choose between ['stand', 'hit']. Your response should be a valid json file in the following format:
{
"thoughts": "{first describe your total points and the dealer's total points then think about which action to choose}",
"action": "stand" or "hit"
}

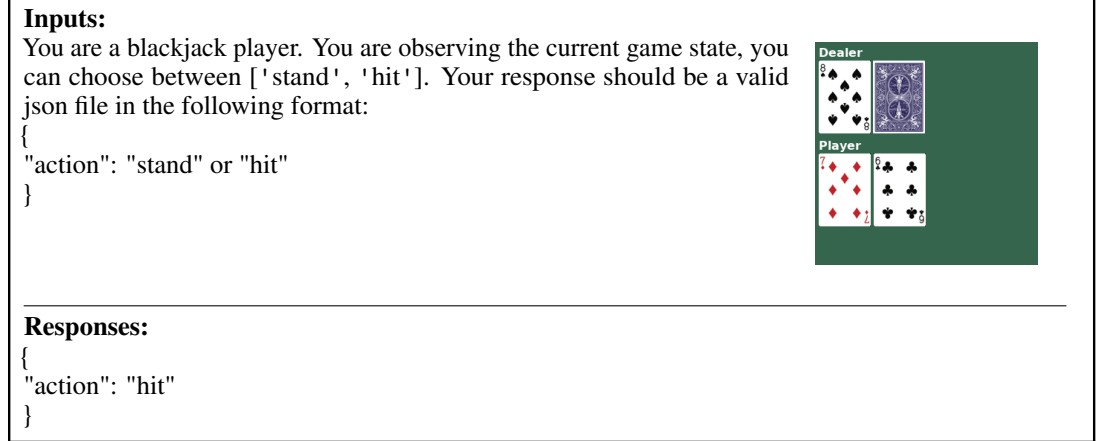

---

**Responses:**

{
"thoughts": "I have 13 points and the dealer has 8 points. I think I should hit.",
"action": "hit"
}

Figure 24: An example of the supervised fine-tuning data for `Blackjack` with CoT.

---

**Inputs:**

You are a blackjack player. You are observing the current game state, you can choose between ['stand', 'hit']. Your response should be a valid json file in the following format:
{
"action": "stand" or "hit"
}

---

**Responses:**

{
"action": "hit"
}

Figure 25: An example of the supervised fine-tuning data for `Blackjack` without CoT.

---

[11]https://www.cs.cmu.edu/~chengwen/blackjack/blackjack.html

## D.3 EZPoints

In the `EZPoints` task, we directly collect 50k instruction-following expert data samples using a task solver. See examples with and without CoT in Figure 26 and 27, respectively.

---

**Inputs:**

You are an expert card game player. You are observing two cards in the image. You are observing the current formula: '5'. You can choose between ['1', '2', '3', '4', '5', '6', '7', '8', '9', '10', '+', '*', '=']. The number or operator you choose will be appended to the current formula. Note that 'J', 'Q', and 'K' count as '10'. Your goal is to output a formula that evaluates to 12, and each number can only be used once. Your response should be a valid json file in the following format:
{
"cards": [x, y],
"current formula": '5',
"thoughts": {First check whether the current formula 'z' is complete. If the current formula 'z' is complete, output '='. Otherwise consider which number or operator should be appended to the current formula to make it equal 12.}
"action": "{number}" or "{operator}"
}

---

**Responses:**

{
"cards": [7, 5],
"formula": "5",
"thoughts": "'5' is an incomplete formula, since '5+7=12', I should append '+' to the current formula", "action": "+"
}"

Figure 26: An example of the supervised fine-tuning data for `EZPoints` with CoT.

---

**Inputs:**

You are an expert card game player. You are observing two cards in the image. You are observing the current formula: '5'. You can choose between ['1', '2', '3', '4', '5', '6', '7', '8', '9', '10', '+', '*', '=']. The number or operator you choose will be appended to the current formula. Note that 'J', 'Q', and 'K' count as '10'. Your goal is to output a formula that evaluates to 12, and each number can only be used once. Your response should be a valid json file in the following format:
{
"action": "{number}" or "{operator}"
}

---

**Responses:**

{
"action": "+"
}"

Figure 27: An example of the supervised fine-tuning data for `Blackjack` without CoT.

## D.4 Points24

In the `Points24` task, we directly collect 50k instruction-following expert data samples using a task solver. See examples with and without CoT in Figure 28 and 29, respectively. Note that the supervised fine-tuning data has a slightly different distribution as the states in the `Points24` environment, since the expert data always have a viable solution to 24, while not all states from `Points24` has a viable solution to 24.

---

**Inputs:**

You are an expert card game player. You are observing two cards in the image. You are observing the current formula: '5'. You can choose between ['1', '2', '3', '4', '5', '6', '7', '8', '9', '10', '+', '*', '=']. The number or operator you choose will be appended to the current formula. Note that 'J', 'Q', and 'K' count as '10'. Your goal is to output a formula that evaluates to 12, and each number can only be used once. Your response should be a valid json file in the following format:

{
"cards": [x, y],
"current formula": '(2',
"thoughts": {First check whether the current formula 'z' is complete. If the current formula 'z' is complete, output '='. Otherwise consider which number or operator should be appended to the current formula to make it equal 12.}
"action": "{number}" or "{operator}"
}

---

**Responses:**

{
"cards": [1, 2, 10, 1],
"formula": "(2",
"thoughts": "'(2' is an incomplete formula, since '(2+10)*(1+1)=24', I should append '+' to the current formula",
"action": "+"
}"

Figure 28: An example of the supervised fine-tuning data for `Points24` with CoT.

---

**Inputs:**

You are an expert card game player. You are observing two cards in the image. You are observing the current formula: '5'. You can choose between ['1', '2', '3', '4', '5', '6', '7', '8', '9', '10', '+', '*', '=']. The number or operator you choose will be appended to the current formula. Note that 'J', 'Q', and 'K' count as '10'. Your goal is to output a formula that evaluates to 12, and each number can only be used once. Your response should be a valid json file in the following format:

{
"action": "{number}" or "{operator}"
}

---

**Responses:**

{
"action": "+"
}"

Figure 29: An example of the supervised fine-tuning data for `Points24` without CoT.

## D.5 ALFWorld

The data collection procedure of the `alfworld` embodied environment is slightly different than the `gym_cards`, as we do not have solvers to generate the instruction-following data, even with the expert text action. Therefore, we take a step back and directly use our prompt as presented in Figure 17 (or Figure 30) to collect 1k episodes (around 45k) instruction-following data from GPT4-V [44] with the CoT reasoning for the instruction-following fine-tuning with CoT. As for the case without CoT, we directly clean the collected CoT by removing the CoT reasonings. See examples with and without CoT in Figure 30 and 31, respectively.

---

**Inputs:**

Your are an expert in the ALFRED Embodied Environment. You are also given the following text description of the current scene: ['You arrive at loc 0. The cabinet 1 is open. On the cabinet 1, you see a pan 1, a kettle 1, a winebottle 1, a apple 1, a stoveknob 1, a stoveknob 2, a stoveknob 3, a stoveknob 4, a knife 1, a saltshaker 1, and a bread 1.']. Your task is to put a cool mug in cabinet. Your admissible actions of the current situation are: ['go to countertop 1', 'go to cabinet 2', 'go to countertop 2', 'go to stoveburner 1', 'go to drawer 1', 'go to drawer 2', 'go to drawer 3', 'go to stoveburner 2', 'go to stoveburner 3', 'go 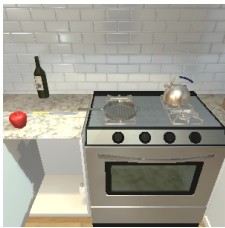 to stoveburner 4', 'go to drawer 4', 'go to cabinet 3', 'go to cabinet 4', 'go to microwave 1', 'go to cabinet 5', 'go to cabinet 6', 'go to cabinet 7', 'go to sink 1', 'go to sinkbasin 1', 'go to fridge 1', 'go to toaster 1', 'go to coffeemachine 1', 'go to cabinet 8', 'go to drawer 5', 'go to drawer 6', 'go to drawer 7', 'go to drawer 8', 'go to shelf 1', 'go to shelf 2', 'go to countertop 3', 'go to shelf 3', 'go to drawer 9', 'go to garbagecan 1', 'open cabinet 1', 'close cabinet 1', 'take pan 1 from cabinet 1', 'take kettle 1 from cabinet 1', 'take winebottle 1 from cabinet 1', 'take apple 1 from cabinet 1', 'take stoveknob 1 from cabinet 1', 'take stoveknob 2 from cabinet 1', 'take stoveknob 3 from cabinet 1', 'take stoveknob 4 from cabinet 1', 'take knife 1 from cabinet 1', 'take saltshaker 1 from cabinet 1', 'take bread 1 from cabinet 1', 'inventory', 'look', 'examine cabinet 1']. Your response should be a valid json file in the following format:
{
"thoughts": "first describe what do you see in the image using the text description, then carefully think about which action to complete the task. ",
"action": "an admissible action"
}

---

**Responses:**

{
"thoughts": "Based on the text description, I am at loc 0 with an open cabinet 1 in front of me. Items such as a pan, kettle, wine bottle, apple, several stoveknobs, a knife, saltshaker, and bread are visible on the cabinet 1. To find a cool mug to put in the cabinet, I may need to first locate a mug, which might be in one of the cabinets or drawers. Since the task does not involve any items currently in cabinet 1, and the mug is not mentioned in the description, my best course of action would be to start by searching the other cabinets or drawers for the mug.",
"action": "go to cabinet 2"
}"

Figure 30: An example of the supervised fine-tuning data for `alfworld` with CoT.

**Inputs:**

Your are an expert in the ALFRED Embodied Environment. You are also given the following text description of the current scene: ['You arrive at loc 0. The cabinet 1 is open. On the cabinet 1, you see a pan 1, a kettle 1, a winebottle 1, a apple 1, a stoveknob 1, a stoveknob 2, a stoveknob 3, a stoveknob 4, a knife 1, a saltshaker 1, and a bread 1.']. Your task is to put a cool mug in cabinet. Your admissible actions of the current situation are: ['go to countertop 1', 'go to cabinet 2', 'go to countertop 2', 'go to stoveburner 1', 'go to drawer 1', 'go to drawer 2', 'go to drawer 3', 'go to stoveburner 2', 'go to stoveburner 3', 'go 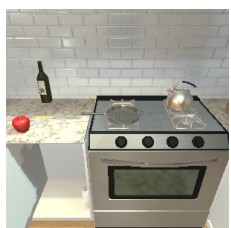 to stoveburner 4', 'go to drawer 4', 'go to cabinet 3', 'go to cabinet 4', 'go to microwave 1', 'go to cabinet 5', 'go to cabinet 6', 'go to cabinet 7', 'go to sink 1', 'go to sinkbasin 1', 'go to fridge 1', 'go to toaster 1', 'go to coffeemachine 1', 'go to cabinet 8', 'go to drawer 5', 'go to drawer 6', 'go to drawer 7', 'go to drawer 8', 'go to shelf 1', 'go to shelf 2', 'go to countertop 3', 'go to shelf 3', 'go to drawer 9', 'go to garbagecan 1', 'open cabinet 1', 'close cabinet 1', 'take pan 1 from cabinet 1', 'take kettle 1 from cabinet 1', 'take winebottle 1 from cabinet 1', 'take apple 1 from cabinet 1', 'take stoveknob 1 from cabinet 1', 'take stoveknob 2 from cabinet 1', 'take stoveknob 3 from cabinet 1', 'take stoveknob 4 from cabinet 1', 'take knife 1 from cabinet 1', 'take saltshaker 1 from cabinet 1', 'take bread 1 from cabinet 1', 'inventory', 'look', 'examine cabinet 1']. Your response should be a valid json file in the following format:

{
"action": "an admissible action"
}

---

**Responses:**

{
"action": "go to cabinet 2"
}"

Figure 31: An example of the supervised fine-tuning data for `alfworld` without CoT.


