# OpenReview forum: "Fine-Tuning Large Vision-Language Models as Decision-Making Agents via Reinforcement Learning"
_NeurIPS.cc/2024/Conference — NeurIPS 2024 poster_

### Official Review · Reviewer_gYxm · 2024-07-05

**Soundness:** 3
**Presentation:** 3
**Contribution:** 2
**Rating:** 6
**Confidence:** 4

**Summary:**

The paper introduces a algorithmic framework that fine-tunes VLM using RL  to enhance their performance in multi-step, goal-directed decision-making tasks. The authors highlight the limitations of traditional visual instruction tuning, which relies on pre-collected datasets and may not effectively train VLMs for interactive decision-making scenarios. Their proposed framework addresses this by providing task descriptions to the VLMs, prompting them to generate chain-of-thought (CoT) reasoning, and translating these into executable actions within an environment. Empirical results show that this method significantly improves the decision-making capabilities of VLMs, enabling them to outperform commercial models like GPT4-V and Gemini.

**Strengths:**

1. The paper is well-written and clear, making it accessible and easy to follow.
2. The proposal of an alternative to current visual instruction tuning for decision-making is intriguing, with well-justified design choices that demonstrate a thoughtful approach.
3. The execution is particularly noteworthy and well-designed. For instance, the method of balancing the influence of chain-of-thought (CoT) output and action output is elegantly handled and demonstrates innovative thinking.

**Weaknesses:**

The paper is good overall. There are a few areas that could benefit from improvement:

1. The current implementation requires fine-tuning for each task individually. However, large Vision-Language Models (VLMs) are generally capable of handling multiple tasks such as Visual Question Answering (VQA) and Optical Character Recognition (OCR).
2. The trade-off between the CoT output and action output is manually set. This means that the balance must be adjusted for different tasks depending on their nature. It is unclear how this parameter can be set for training a multi-task policy.
3. Despite the use of Low-Rank Adaptation (LoRA) for fine-tuning, the computational cost remains high.
4. The framework shows promise, but the evaluation tasks do not fully convey the benefits of using a multi-modal large language model (MLLM). Specifically, the model's ability to reason or the potential to explicitly correct its own behavior in scenarios where this is crucial is not thoroughly demonstrated.

**Questions:**

It would be interesting to explore how visual prompting could enhance the framework and improve performance.

**Limitations:**

yes.

---

> ### Author Rebuttal · Authors · 2024-08-06
>
> Dear reviewer gYxm,
>
> Thank you very much for your positive feedback and insightful suggestions on our paper! We are delighted to hear that you appreciated the presentation of our work and our innovative method!
>
> ---
> ### On the computational cost of LoRA
>
> We’d like to note that the computational costs of our LoRA approach are comparable to standard supervised LoRA fine-tuning. A computationally similar scenario, RLHF, has been extensively studied and proven to work well, even in resource-constrained academic labs. Additionally, by switching between different LoRAs with the same base model, we only maintain one set of VLM weights, reducing VRAM consumption to approximately one-third of standard PPO training. Looking ahead, we are excited about future algorithmic improvements to PPO that will further reduce training costs.
>
> ---
> ### On multi-task training
>
> Thank you for pointing this out. Our current framework only allows training in a single environment but is not limited to a single task. For example, experiments in ALFWorld suggest that our framework can perform well in a multi-task environment containing different types, potentially shedding light on its multi-task learning capabilities (in a single environment) as well.
>
>
> ### Regarding other weaknesses and questions
>
> 1. **On the hyperparameter of the CoT tokens.** We agree that manually setting a hyperparameter to adjust the log probabilities of the CoT tokens is not algorithmically elegant. This could be improved by providing an auto-tuning mechanism to handle the additional hyperparameter. Due to limited development time, we are unable to conduct such additional experiments during the rebuttal period, and we would like to leave this for future studies.
> 2. **On exploiting the capability of MLLM.** We agree that we have only provided a working version of the end-to-end RL training framework, without extensively exploiting the capability of the MLLM. Using our framework, one can extensively try other methods (e.g., different prompting techniques) to exploit the capabilities of the MLLM.
> 3. **On the visual prompting.** Thanks for the suggestion! While our current framework does not naturally allow visual prompting (VP), it is indeed interesting to see the performance of incorporating VP into our framework, we would like to leave this for future study.
>
> ---
> ### Concluding remarks
> We highly appreciate your appreciation of our work, suggestions for improvements, and potential future studies! Please be assured that we will include an additional paragraph to discuss the computational cost compared to RLHF, and the multi-task training capabilities in the updated version. Although we are only able to address 2 out of your 4 concerns, **if you think addressing these 2 concerns or our additional experiments further improve the quality of our work, would you mind kindly improving the rating of our work?** However, if you feel this is not sufficient, we understand, as the other suggestions you made are already very important future directions.

---

> > ### Comment · Reviewer_gYxm · 2024-08-12
> >
> > Thank you for your response. I have no further questions at the moment. While I remain positive about the paper, I do agree that Reviewer MKrq's point is valid. A deeper exploration of the role of reinforcement learning in the context of the paper could be both interesting and important.

---

> > > ### Author Response · Authors · 2024-08-12
> > >
> > > Thank you for the follow-up and your support. We are actively studying the effect of RL for follow-up work, please stay tuned!

---

### Official Review · Reviewer_QEzb · 2024-07-10

**Soundness:** 3
**Presentation:** 3
**Contribution:** 3
**Rating:** 7
**Confidence:** 4

**Summary:**

This paper provides a framework that fine-tunes a large vision-language model (VLM) with reinforcement learning (RL) for decision-making tasks requiring vision and language understanding.

In the framework, the VLM takes as input a state s_t that contains a visual observation o_t and input prompt v_t^{in}. The input prompt v_t^{in} contains a task description, legal action space, and desired output. Then, the VLM generates text v_t^{out} that contains CoT reasoning and an open-ended text action. The framework applies a post-processing function f on the open-ended text action, and obtain a legal action a_t. Then, the framework inputs the legal action a_t to the environment and obtains reward r(s_t, a_t) and the next visual observation o_{t+1}. Based on the reward r(s_t, a_t), the framework fine-tunes the VLM with RL (i.e., PPO).

The proposed framework is evaluated on two environments: gym_cards and ALFWorld. The gym_cards environment contains four tasks like NumberLine, EZPoints, Points24, and Blackjack, and these tasks require an ability to recognize numbers in figures for arithmetic reasoning. This paper empirically demonstrates that the LLaVA-7B model fine-tuned by the proposed framework can outperform close-source LLMs such as GPT4-V and Gemini on these environments.

**Strengths:**

S1. This paper provides a method that can fine-tune VLMs with RL for decision-making tasks. The overall problem setting seems novel and interesting.

S2. Technically, the method consists of (1) prompt design for domain-specific outputs, (2) post-processing open-ended text for legal actions, and (3) estimating action probabilities of VLM policies. Among them, the third technique is very interesting. To generate a better text action, the VLM generates CoT reasoning text before it. To estimate action probabilities of VLM policies, this paper proposes to split the probability of generating CoT reasoning and the probability of generating a text action.

S3. This paper empirically demonstrates that LLaVA-7B model fine-tuned with RL can outperform GPT-4V and Gemini on the gym-card environment that requires vision and language understanding.

**Weaknesses:**

W1. Even though this paper demonstrates that LLaVA-7B fine-tuned with RL outperforms GPT4-V on the gym-card environment, according to Table 2, on ALFWorld, the performance gain does not seem significant (GPT4-V 19.4 vs Ours 21.7).

**Questions:**

Q1. Regarding the weaknesses W1 above, is there any reason the performance gain does not be significant on ALFWorld?

**Limitations:**

L1. The authors provide limitations in the Section 7 (i.e., Conclusions, Limitations, and Future Directions).

---

> ### Author Rebuttal · Authors · 2024-08-06
>
> Dear reviewer QEzb,
>
> Thank you very much for your high appreciation of our work, we are glad to hear that you appreciated the novelty, technicality, and performance of our paper!
>
> ---
>
> ### Regarding the explanation of the limited performance on ALFWorld
>
> Besides your appreciation, we would like to provide some potential explanations for the limited performance of the ALFWorld environment as you have mentioned in the weakness and question sections. The ALFworld environment is generally harder than the gym_card environment in these two following aspects:
>
> 1. Each game in the `gym_card` is a single-task environment, while the `ALFworld` is a **multi-task environment**, which substantially increases the difficulty for RL training in `ALFworld`, as the agent needs to learn more diverse actions based on the types of tasks. In fact, showing improvements in the `ALFworld` environment also sheds light on the multi-task learning capabilities of our framework.
> 2. In addition, the action space of `ALFworld` is generally much larger than the action space in gym_cards, which makes the learning process of `ALFworld` more difficult. More specifically, see Figure 17 on page 19 for example: (1) the agent is required to choose from more than 40 admissible actions at every step; (2) and at each step the admissible action set may be different.
>
> ---
> ### Concluding remarks
>
> We highly appreciate your suggestion for the additional discussion on the limited performance of the `ALFWorld`, and we will definitely include them in the discussion section. **If you think our response addresses your concerns or our additional experiments improve the quality of this work, would you mind kindly further improving your rating? If not, please let us know as well, and we would like to further engage and improve our work based on your future suggestions!**  Thank you again for your appreciation and your insightful suggestions for improving our paper!

---

> ### Comment · Reviewer_QEzb · 2024-08-13
> **After the Author Response**
>
> Thank you for providing a thoughtful response to my question. I could understand this paper more. I maintain my initial rating. By the way, if the authors will give a name to the proposed method, it will be easier for readers to refer to it. :-)

---

> > ### Author Response · Authors · 2024-08-13
> >
> > Thank you for the kind suggestion on the name of the paper :) we tried some names but never found a suitable one. We will try harder and hopefully we can find one for the updated version.
> >
> > Thanks again for your suggestions and your appreciation!

---

### Official Review · Reviewer_RYx1 · 2024-07-12

**Soundness:** 3
**Presentation:** 3
**Contribution:** 3
**Rating:** 7
**Confidence:** 4

**Summary:**

- This paper studies the training of vision-language models for decision-making tasks via reinforcement learning.
- The authors train a 7B parameter Llava model and a baseline CNN model with proximal policy optimization (PPO) on the alfworld and `gym_cards` environments.
- Additionally, the authors study the impact of also generating and reinforcing CoT rationales produced by the VLM.
- They find that RL is often able to substantially improve the performance of a fine-tuned VLM, and that producing CoT tokens significantly helps during exploration.

**Strengths:**

- This is a useful empirical paper that serves as a sanity check for using RL on multimodal tasks.
- I was surprised to see the impact that reinforcing CoT tokens had. I didn't expect this. This could be a useful signal for the community.
- The experimental setup is sound and well-designed. I especially liked the carefully prepared Llava-sft baseline.
- The paper makes a convincing case that RL adds something beyond SFT, at least for `gym_cards`.

**Weaknesses:**

- Only one VLM. I would like to have seen whether the conclusions change (even on a single environment) with a different VLM or different sizes of VLM.

**Questions:**

- I would have liked to see more of an explanation of why RL does not outperform SFT across the board on Alfworld.

**Limitations:**

Not applicable.

---

> ### Author Rebuttal · Authors · 2024-08-07
>
> Dear reviewer RYx1,
>
> Thank you very much for your high appreciation of our work. We are delighted to hear that you found our results in CoT and SFT insightful.
>
> ---
> ### General response
>
> In addition to your appreciation, we plan to incorporate the following results and discussions into the updated paper based on your valuable suggestions and questions:
> 1. **We have conducted additional experiments on the recently released [Cambrian-1](https://github.com/cambrian-mllm/cambrian) model, which has significantly better vision recognition capabilities.** Detailed results are provided below.
> 2. **We will include a more comprehensive explanation of the discussion on RL and SFT in Alfworld**, with further details also provided below.
>
> ---
> ### Experiments on additional VLM
> Thank you for your suggestion. We conducted additional experiments using the recently released [Cambrian-1](https://github.com/cambrian-mllm/cambrian), a VLM with enhanced visual capabilities, on the EZPoints and Points24 tasks. We provide all experiments using the same setup in the paper but with Cambrian-1-8b as our backbone VLM.
> | Base VLM                        | EZP | P24   |
> |---------------------------------|-----|-------|
> | Cambrian-1-8b                   | 54.0% | 9.1%   |
> | Llava-1.6-7b (same in paper)    | 50.0% | 2.3%  |
>
>
> As shown in the table above, the final performance improves when the backbone VLM has better visual capabilities (`EZP: 50.0% -> 54.0%`, `P24: 2.3% -> 9.1%`), which further demonstrates that (1) our framework can adapt to other VLMs as well; (2) VLMs with better visual capabilities will enjoy better final performance. **Note that we have not extensively swept the hyperparameters on Cambrian-1 due to the limited time, we believe the performance of Cambrian-1 can be further improved after more parameter sweeping.**
>
> To see how Cambrian-1 is better than Llava-1.6 quantitatively, we ran additional experiments to compare the number recognition accuracy in `EZP` and `P24`. More specifically, we evaluate the accuracy of the original llava-1.6-7b / Cambrian-1-8b checkpoints (0-shot) and the supervised fine-tuned (SFT) checkpoint. The results below are tested with the same prompt in the paper and averaged among 1000 trials. Each trial is considered correct, if *all* numbers (or cards) are correctly recognized. For example, in the EZpoint and Points24, a trial is considered correct if the VLM recognizes the numbers in *all* cards.
>
> | Recognition Accuracy | Llava 0-shot | Llava SFT | Cambrian 0-shot | Cambrian SFT |
> |----------------------|--------------|-----------|-----------------|--------------|
> | EZP                  | 0.10         | 0.79      | 0.92         |     0.98     |
> | P24                  | 0.01         | 0.48      |  0.70        |     0.73      |
>
>
> **In conclusion, we showed that (1) our framework can still improve the performance of another VLM (Cambrian-1-8b); (2) and using Cambrian-1-8b, a mode with better visual capability, as the backbone will achieve much better performance compared to Llava.**
>
> ---
> ### Explanations on why RL does not improve across all the SFT tasks in ALFworld
>
> Thank you very much for raising this insightful question. One potential explanation for this phenomenon is that the `ALFWorld` is a multi-task environment, where each task in `gym_cards` is trained in a single-task environment, which makes `ALFWorld` substantially harder than tasks in `gym_cards`. Following this line of thought, it would also be interesting to explore how RL can improve the multi-task learning capability of VLMs using the proposed framework. We sincerely thank the reviewer for raising this intriguing question, and we will include it in the discussion section in our updated draft.
>
> ---
> ### Concluding remarks
> We highly appreciate your suggestion for additional experiments on another VLM and discussions on the performance of ALFworld. We will definitely include them in the discussion section in the future update. **If you think our additional experiments and discussion address your concerns, would you mind kindly further improving your rating?** Thank you again for your appreciation and your valuable suggestions for improving our paper!

---

> > ### Comment · Reviewer_RYx1 · 2024-08-13
> >
> > I have read the response. I will maintain my original rating of a 7. This paper should be accepted. It presents an empirical study with some interesting and potentially useful observations, but the empirical testbed is limited, so the domain of applicability of the observations is unclear.

---

> > > ### Author Response · Authors · 2024-08-13
> > >
> > > Thank you for your support and acknowledgment!

---

### Official Review · Reviewer_MKrq · 2024-07-30

**Soundness:** 2
**Presentation:** 3
**Contribution:** 2
**Rating:** 3
**Confidence:** 4

**Summary:**

The paper proposes an algorithmic framework for fine-tuning large vision-language models (VLMs) using reinforcement learning (RL) for multi-step decision-making tasks. The framework enhances VLMs' reasoning and decision-making capabilities by incorporating chain-of-thought (CoT) reasoning. The empirical results demonstrate that the proposed method outperforms existing models like GPT-4V and Gemini in various decision-making tasks, highlighting the significance of CoT reasoning.

**Strengths:**

1. The topic is interesting and valuable to the community.
2. The presentation is clear, and the paper is easy to follow.
3. The methodology is general sound.

**Weaknesses:**

My primary concerns lie with the experimental results and methodology. Please refer to the questions below for detailed points.

**Questions:**

1. In the appendix, it is observed that CoT not only provides thought processes but also manually encoded key states, such as the “current number” and “target number” in the NumberLine task (Figure 9). This crucial information is hidden in the main text (See Figure 3). The authors need to clearly differentiate the source of CoT's effectiveness in improving reasoning—whether it stems from manually labeled key states or the textual thought process itself. This distinction should be experimentally validated and presented in the main text rather than being relegated to the appendix.

2. Moreover, regarding the CoT issue, Figures 11 and 13 show that CoT encodes not only the key states used for decision-making but also information "directly obtained from the formula in the image." There seems to be no reasonable justification for this, as theoretically, this information can be derivable from the images. An effective RL method should be capable of optimizing to identify this critical information through rewards. The inclusion of this manually encoded information makes the experiment tricky because it allows any difficult image understanding problem to be reduced to a manual extraction process. Authors should not manually encode such information to boost the final performance of their method.

3. Did your baselines (GPT4-V and Gemini) include CoT information?

4. To validate the effectiveness of the proposed RL algorithm, it would be beneficial to compare it with other RL methods, such as "OFFLINE RL FOR NATURAL LANGUAGE GENERATION WITH IMPLICIT LANGUAGE Q LEARNING" and "ArCHer: Training Language Model Agents via Hierarchical Multi-Turn RL." If using the authors' SFT model with your CoT, can we replace the proposed RL algorithm with these RL methods to achieve optimization?

---

> ### Author Rebuttal · Authors · 2024-08-06
>
> Dear reviewer MKrq,
>
> Thank you very much for your valuable review and your questions! We will definitely integrate them into the updated paper to address your concerns.
>
> ---
> ### General response
>
> We sincerely thank you for your appreciation of our work and your insightful suggestions to improve our paper. The primary contribution of our work is to **provide the first integrated system that enables end-to-end RL fine-tuning on VLMs, incorporating customized prompts as inputs**. Hence, we directly utilized PPO, which has been well studied in RLHF, rather than comparing the effects of different prompts or RL algorithms within our framework, as these topics are not the primary focus of our paper. However, we highly value your suggestions regarding the effects of different prompts and comparisons of various RL algorithms. These are indeed important topics for future studies within our framework, and we will incorporate more discussions on these subjects as future work in the updated paper draft. Once again, we greatly appreciate your attention to detail and your thoughtful feedback.
>
> ---
> To answer your questions:
>
>
> ## Q.1:
>
> ### Regarding the presentation in Figure 3.
> > In the appendix, it is observed that CoT not only provides thought processes but also manually encodes key states, such as the “current number” and “target number” in the NumberLine task (Figure 9). This crucial information is hidden in the main text (See Figure 3).
>
> A: Thank you for bringing to our attention the presentation issue in Figure 3. We appreciate your careful review. The main purpose of Figure 3 is to provide a template for our expected input and output, for facilitating the post-processing described in Section 4.2, rather than detailing the design of each task-specific prompt. We apologize for any confusion this may have caused. To improve clarity, we will update the figure to include an additional key, "{optional task-specific prompts}," before "thoughts" to elaborate on the difference in our task-specific prompts carefully.
>
>
> > The authors need to clearly differentiate the source of CoT's effectiveness in improving reasoning—whether it stems from manually labeled key states or the textual thought process itself. This distinction should be experimentally validated and presented in the main text rather than being relegated to the appendix.
>
>
> A: Once again, we apologize for any confusion in our presentation and thank the reviewer for highlighting this point. Our intention is not to compare the performance of different prompts, but to **present an end-to-end RL fine-tuning paradigm that effectively utilizes customized CoT prompting for VLM**. However, we agree that studying the effects of different customized prompts using intermediate reasoning steps [1,2] could potentially further improve the performance, and we would like to leave that for future research.
>
> ## Q.2:
> > Moreover, regarding the CoT issue, Figures 11 and 13 show that CoT encodes not only the key states used for decision-making but also information "directly obtained from the formula in the image." There seems to be no reasonable justification for this, as theoretically, this information can be derivable from the images. An effective RL method should be capable of optimizing and identifying this critical information through rewards.
>
> A: We sincerely thank you for your valuable suggestions to further improve the presentation of our work. We agree that the formula can be directly obtained from the image in principle; however, the backbone VLM (Llava-1.6b) struggles to recognize the formula from the images, which causes the entire pipeline to fail.
>
>
> To demonstrate this, we provide additional evaluation results on the visual recognition capabilities of EZPoints, and Points24 in the Table below. In the table below, we have conducted additional experiments to evaluate the accuracy of the visual recognition for cards and equations in EZPoints and Points24, on the original llava-1.6-7b checkpoint (0-shot) and the supervised fine-tuned (SFT) checkpoint.
>
> The results in the table below are tested with the same prompt in the paper and averaged among 1000 trials. For recognizing numbers, each trial is considered correct, if *all* numbers (or cards) are correctly recognized. For example, in the EZpoint and Points24, a trial is considered correct if the VLM recognizes the numbers in *all* cards. For recognizing equations, each trial is considered correct if the VLM successfully recognizes the equation in the image.
>
>
> | Recognition Accuracy | 0-shot          |               | SFT            |               |
> |----------------------|-----------------|---------------|----------------|---------------|
> |                         | Numbers         | Equation      | Numbers        | Equation      |
> | EZP                 | 0.10            | 0.07          | 0.79           | 0.12          |
> | P24                 | 0.01            | 0.03          | 0.48           | 0.03          |
>
>
>
> As shown in the table above, while the VLM can reasonably recognize the cards or numbers from the images, it encounters significant difficulties in recognizing the equations – even for EZPoint, the accuracy of recognizing equations is roughly 12% after supervised fine-tuning. Hence, if we ask the VLM to recognize both the numbers and the equation at the same time, the probability of the agent being able to recognize both the equation and the cards throughout the entire trajectory becomes very low – e.g., in the EZPoint environment, the probability of the agent that can successfully recognize the equations for at least 3 steps, is less than $0.12^3\approx0.0017$.
>
> **Page 1/ 2, to be continued.**

---

> ### Author Response · Authors · 2024-08-06
> **Follow up rebuttal page**
>
> **Continued from page 1.**
>
> Therefore, our framework cannot provide any performance gain due to the limited visual capability of the backbone VLM. We have also briefly discussed such failure examples in Appendix B.5, where our framework fails when the backbone VLM is unable to recognize the key information for the task.
>
> Please be assured that we will include the aforementioned table and an additional discussion on how the visual capabilities of the backbone VLM affect the final performance of our method in the updated version. We appreciate your suggestion for improving the presentation of our paper.
>
> ## Q.3:
> > Did your baselines (GPT4-V and Gemini) include CoT information?
>
> A: Thank you for the question. Yes, our experimental baselines with GPT4-V and Gemini use the same task-specific prompt for each task, as we discussed in line 711 of Appendix B.2.
>
> ## Q.4:
> > To validate the effectiveness of the proposed RL algorithm, it would be beneficial to compare it with other RL methods, such as "OFFLINE RL FOR NATURAL LANGUAGE GENERATION WITH IMPLICIT LANGUAGE Q LEARNING" and "ArCHer: Training Language Model Agents via Hierarchical Multi-Turn RL." If using the authors' SFT model with your CoT, can we replace the proposed RL algorithm with these RL methods to achieve optimization?
>
> A: Once again, we sincerely thank the reviewer for suggesting additional comparison experiments with ILQL [3] and ArCHer [4]. We agree that our end-to-end RL training framework can, in principle, adopt other RL algorithms besides PPO. Studying how different RL algorithms affect VLM training is indeed an important topic for future research.
> However, due to the limitations of computational resources and the nature of training large models, we are unable to provide results for more RL algorithms on VLMs during the rebuttal period. As mentioned in footnote 2 on page 6, each curve in our experiment takes 30 hours on 4 A100 GPUs to run. This time estimate only covers running the experiments and does not include the integration of a new RL algorithm into VLM training, which generally requires significantly longer development time (e.g., for this project, we spent more than 2 months developing the framework just for PPO). Similar computation cost issues have also been discussed in [3,4]. In section 5.7, section 6 of [4], and section 7 of [3], computational budget is a major bottleneck for comparing different RL algorithms on large model training. Moreover, our backbone VLM (llava-1.6-7b) is substantially larger than the backbone models used in ArCHer and ILQL. For example, ArCHer conducted most experiments on GPT-2, a 1.5b model (Section 6 in [4]), and ILQL conducted all experiments on GPT-2 (Appendix A.4 in [3]).
> We truly appreciate the reviewer’s suggestion to compare different RL algorithms within our framework, and we will definitely update our paper by discussing the comparison with different algorithms in future work.
>
> ---
> ### Concluding remarks
> We would like to thank the reviewer again for the insightful suggestions for improving our paper. Please be assured that we will update our paper to better (1) clarify the usage of our task-specific prompts in Figure 3; (2) add more discussion on the computational limitation for comparing with other RL algorithms; and (3) include the additional evaluation results of the visual recognition accuracy of llava-1.6-7b as presented before.
>
> **Please let us know if our response addresses your concerns, if so, would you mind kindly improving your rating for our work? If not, please let us know as well, and we would like to further engage and improve our work based on your future suggestions!**
>
> ---
> [1] Wei, Jason, et al. "Chain-of-thought prompting elicits reasoning in large language models." Advances in neural information processing systems 35 (2022): 24824-24837.
>
> [2]  Fu, Yao, et al. "Complexity-based prompting for multi-step reasoning." The Eleventh International Conference on Learning Representations. 2023.
>
> [3] Snell, Charlie, et al. "Offline rl for natural language generation with implicit language q learning." The Eleventh International Conference on Learning Representations. 2023.
>
> [4] Zhou, Yifei, et al. "Archer: Training language model agents via hierarchical multi-turn rl." ICML, 2024.
>
> **Page 2/2.**

---

> ### Comment · Reviewer_MKrq · 2024-08-11
>
> I would like to appreciate the authors’ faithful rebuttal. Here is my follow-up response:
>
> > "Our intention is not to compare the performance of different prompts, but to present an end-to-end RL fine-tuning paradigm that effectively utilizes customized CoT prompting for VLM. However, we agree that studying the effects of different customized prompts using intermediate reasoning steps [1,2] could potentially further improve the performance, and we would like to leave that for future research."
>
> I do not mean to study different customized prompts. Mentioning "different customized prompts" is because your current implementation is problematic in this regard. When stating “We provide the first integrated system that enables end-to-end RL fine-tuning on VLMs”, at least we expect that the RL fine-tuning techniques enhance the visual understanding capabilities. Otherwise, it can be just a fine-tuning technique for LLMs. Unfortunately, I cannot find solid evidence in the current experiments to support this claim.
>
> > However, the backbone VLM (Llava-1.6b) struggles to recognize the formula from the images, which causes the entire pipeline to fail.
>
> It is acceptable to report failed results.  If your baseline also excludes this manually extracted information, they are under fair comparison. However, keep the information in the experiments gives the community a false sense of success and burdens future researchers who aim to solve this problem directly.
>
> Suggestions: After removing those manually extracted information, perhaps the authors could demonstrate that RL training improves the accuracy of information extracted by VLMs from images. This would provide direct evidence that this is a viable RL training method for VLMs.

---

> > ### Author Response · Authors · 2024-08-11
> >
> > We thank the reviewer for the follow-up. We would like to make the following clarifications:
> >
> > > When stating “We provide the first integrated system that enables end-to-end RL fine-tuning on VLMs”, at least we expect that the RL fine-tuning techniques enhance the visual understanding capabilities.
> >
> > We never claimed our method **improves the performance of the visual understanding capabilities of the VLM**, our main contribution in this work is **to provide a post-training framework for improving VLM’s decision-making capabilities** (e.g., see line 43 of our introduction). Such performance improvement is well justified by our experiments while comparing to SFT, GPT4-v, and Gemini **with the same prompt**. The performance improvement could come from these different possibilities (1) RL can improve VLM’s visual understanding capabilities; (2) RL can improve VLM’s language reasoning capabilities, or (3) both, and we thank you for mentioning one possibility for the performance improvement.
> >
> > >  Otherwise, it can be just a fine-tuning technique for LLMs. Unfortunately, I cannot find solid evidence in the current experiments to support this claim.
> >
> > The current VLMs (Llava, GPT4-v, Gemini) actually allow pure text inputs as well, and in principle, our method can be applied to a pure text environment without using the vision input as well. We believe the applicability of our post-training technique on LLM should be a further strength of our paper rather than a weakness.
> >
> >
> > > It is acceptable to report failed results. If your baseline also excludes this manually extracted information, they are under fair comparison.
> >
> > For each task, our experiments on other VLMs (llava-sft, Gemini, GPT4-v) are using the same prompt (with or without the prompt for key information). Hence we believe such a comparison should be fair.
> >
> > > However, keep the information in the experiments gives the community a false sense of success and burdens future researchers who aim to solve this problem directly.
> >
> > On the contrary, we believe that our framework works better with customized CoT prompting and actually provides more information to the community – since it also suggests that CoT significantly improves the decision-making capabilities [1,2] of VLMs under our framework as well, we have also extensively studied this in Section 6.2 on the effect of the CoT prompts.
> >
> > > Suggestions: After removing the manually extracted information, perhaps the authors could demonstrate that RL training improves the accuracy of information extracted by VLMs from images. This would provide direct evidence that this is a viable RL training method for VLMs.
> >
> > Thank you for your suggestion, we agree this is an important question to study in the future, under the hypothesis that **Can RL improve the visual understanding capabilities of the VLMs**, which is **different from** the main hypothesis of this work (**can RL improve VLM’s decision-making capabilities**). Still, this is a very interesting direction and we appreciate the reviewer for pointing it out.
> >
> > **Please let us know if you have any other concerns or suggestions, we would be happy to engage further!**
> >
> > [1] Wei, Jason, et al. "Chain-of-thought prompting elicits reasoning in large language models." Advances in neural information processing systems 35 (2022): 24824-24837.
> >
> > [2]  Fu, Yao, et al. "Complexity-based prompting for multi-step reasoning." The Eleventh International Conference on Learning Representations. 2023.

---

> > > ### Comment · Reviewer_MKrq · 2024-08-12
> > >
> > > Thanks for the authors' detailed response and the efforts made in the rebuttal. At this point, I maintain my original concerns; however, I am open to reconsidering my evaluation based on the suggestions from the Area Chairs.

---

> > > > ### Author Response · Authors · 2024-08-12
> > > >
> > > > Thank you for the follow-up. We hope you have a nice rest of your week.

---

### Author Rebuttal · Authors · 2024-08-07

Dear reviewers,

We would like to express our sincere gratitude for your overall positive recommendations and insightful suggestions regarding our work. Based on the overall feedback, **we have conducted two additional experiments**, which we believe further enhance the quality of our work. Specifically, we have added:

1. **An evaluation of the visual recognition capabilities of the backbone VLM (Llava-1.6-7b) on EZPoints, and Points24**, illustrating that the visual recognition capabilities of the backbone VLM are a major bottleneck for further performance improvement.
2. **Additional experiments using the recently released [Cambrian-1](https://github.com/cambrian-mllm/cambrian) model**, which has better visual capabilities, to demonstrate that (1) our framework can work with another VLM (as suggested by reviewer RYx1), and (2) with improved visual capabilities, task performance can be further improved.

---
### Additional evaluation results on the visual capabilities of the backbone VLM (Llava-1.6)
In the table below, we present the results of additional experiments evaluating the accuracy of visual recognition for cards and equations in EZPoints and Points24, using the original Llava-1.6-7b checkpoint (0-shot) and the supervised fine-tuned (SFT) checkpoint. The results are based on 1000 trials and tested with the same prompt used in the paper. For number recognition, each trial is considered correct if all numbers (or cards) are correctly recognized. in EZPoints and Points24, a trial is correct if the VLM recognizes the numbers on all cards. For equation recognition, each trial is correct if the VLM successfully recognizes the equation in the image.


| Recognition Accuracy | 0-shot          |               | SFT            |               |
|----------------------|-----------------|---------------|----------------|---------------|
|                         | Numbers         | Equation      | Numbers        | Equation      |
| EZP                 | 0.10            | 0.07          | 0.79           | 0.12          |
| P24                 | 0.01            | 0.03          | 0.48           | 0.03          |


As shown in the table above, the number recognition accuracy of our current backbone VLM (Llava-1.6-7b) decreases (`0.10 -> 0.01`) when the task difficulty increases (`EZP -> P24`). While the supervised fine-tuning can improve the accuracy of all tasks: `EZP` (`0.10 -> 0.79`), and `P24` (`0.01 -> 0.48`), the backbone VLM still suffers from recognizing the equations from the images, even after supervised fine-tuning.

**In conclusion, these evaluation results suggest that the backbone VLM suffers from poor visual recognition**, which can be slightly improved by supervised fine-tuning, but SFT does not always enhance visual capabilities across all tasks.

---
### Additional experiments using another backbone VLM (Cambrian-1)
To evaluate the adaptability of our framework with another VLM, **we conducted additional experiments using the recently released [Cambrian-1](https://github.com/cambrian-mllm/cambrian), a VLM with enhanced visual capabilities, on the EZPoints, and Points24 tasks as the backbone VLM for our framework**. The experiments were conducted using the same setup as in the paper but with Cambrian-1-8b as the backbone VLM.
| Base VLM                        | EZP | P24   |
|---------------------------------|-----|-------|
| Cambrian-1-8b                   | 54.0% | 9.1%  |
| Llava-1.6-7b (same in paper)    | 50.0% | 2.3%  |

As shown in the table above, the final performance improves when the backbone VLM has better visual capabilities (`EZP: 50% -> 54%`, `P24: 2.3% -> 9.1%`), which further demonstrates that (1) our framework can adapt to other VLMs as well; (2) VLMs with better visual capabilities will enjoy better final performance. **Note that we have not extensively swept the hyperparameters on Cambrian-1 due to the limited time, we believe the performance of Cambrian-1 can be further improved after more parameter sweeping.**
To quantify how Cambrian-1 compares to Llava-1.6, we ran additional experiments to evaluate number recognition accuracy in `EZP` and `P24`. Specifically, we evaluated the accuracy of the original Llava-1.6-7b / Cambrian-1-8b checkpoints (0-shot) and the supervised fine-tuned (SFT) checkpoint. The results are based on 1000 trials, using the same prompt as in the paper. Each trial is considered correct if all numbers (or cards) are correctly recognized.

| Recognition Accuracy | Llava 0-shot | Llava SFT | Cambrian 0-shot | Cambrian SFT |
|----------------------|--------------|-----------|-----------------|--------------|
| EZP                  | 0.10         | 0.79      | 0.92         |     0.98     |
| P24                  | 0.01         | 0.48      |  0.70        |     0.73      |



**In conclusion, we showed that (1) our framework can improve the performance of another VLM (Cambrian-1-8b), and (2) using Cambrian-1-8b, a model with better visual capability, as the backbone achieves much better performance compared to Llava.**

---

### Decision · Program_Chairs · 2024-09-25

**Decision:**

Accept (poster)

**Comment:**

This work introduces a novel approach using reinforcement learning to enhance the decision-making capabilities of vision-language models (VLMs). Three reviewers have expressed positive feedback, while one reviewer has raised some concerns. The Area Chair finds the work interesting and recommends it for publication at NeurIPS 2024. The reviewers have highlighted some valuable concerns that should be addressed in the final camera-ready version. The authors are encouraged to incorporate the necessary revisions in the final submission.